# Impact of deforestation and temporal land-use change on soil organic carbon storage, quality, and lability

**Emmanuel Amoakwah**[1,2], **Shawn T. Lucas**[3], **Nataliia A. Didenko**[4], **Mohammad A. Rahman**[2], **Khandakar Rafiq Islam**[2]*

**1** Council for Scientific and Industrial Research (CSIR)–Soil Research Institute, Kwadaso-Kumasi, Ghana, **2** Soil, Water, and Bioenergy Resources, The Ohio State University South Centers, Piketon, Ohio, United States of America, **3** College of Agriculture, Community, and the Sciences, Kentucky State University, Frankfort, Kentucky, United States of America, **4** Institute of Water Problems and Land Management, Kyiv, Ukraine

* islam.27@osu.edu

**Data Availability Statement:** All relevant data are within the paper and its Supporting Information files.

## Abstract

Soil organic carbon (SOC) plays a key role in regulating soil quality functions and ecosystem services. The objective of our study was to evaluate the impact of deforestation and subsequent land-use change on the SOC and total nitrogen (TN) concentration, quality, and lability under otherwise similar soil and environmental conditions. Geo-referenced composite soils (0 to 30 cm depth at 7.5 cm interval) sampled from agriculture, bioenergy plantations (*Miscanthus* x *giganteus*), Conservation Reserve Program (CRP), and wetland were analyzed for SOC, TN, active C (AC), humic- and fulvic acid (HA and FA), non-humic C (NH), $E_4$: $E_6$ ratio, humification indices (HI, HR, and DH), and carbon and nitrogen management indices (CPI, NPI, and CMI), compared to soils under protected forest as a control. Results showed that the CRP had the highest depth distribution and profile-wise stocks of SOC, TN, AC, and FA with respect to the lowest in agriculture upon conversion of forest. Moreover, the SOC and TN contents were significantly stratified in the CRP when compared to agriculture. While agriculture had the wider HA: FA ratios with highest HI and HR but lowest DH values, the CRP, in contrast, had the narrow HA: FA ratios with lowest HI but highest DH values, when compared to the forest. Spectral analyses have shown lower $E_4$: $E_6$ ratios under the forest when compared to both agriculture and the CRP; however, the later had significantly higher $E_4$: $E_6$ ratios than that of agriculture. The CPI, as measures of SOC accumulation or depletion, significantly decreased by 16% under agriculture but increased by 12% under the CRP. While the CMI, as measures of SOC accumulation or depletion and lability, with higher values under the CRP suggested a proportionally more labile SOC accumulation, in contrast, the smaller values under agriculture indicated a greater depletion of labile SOC over time. Moreover, the CRP may have favored a more labile SOC accumulation with higher proportions of aliphatic C compounds, whereas agriculture may have a SOC with high proportions of non-labile aromatic C compounds. Principal components analysis clearly separated and/or discriminated the land-use impacts on soil carbon pools and TN. Likewise, redundancy analysis of the relationship between measured soil parameters and

**Funding:** This work was funded by the Norman Borlaug Leadership Enhancement in Agriculture Program (Borlaug LEAP) through a grant to the University of California-Davis by the United States Agency for International Development (USAID). There was no additional external funding received for this study. The funders did not play any role in the study design, data collection and analysis, decision to publish, or preparation of the manuscript.

**Competing interests:** The authors have declared that no competing interests exist.

land-use validated that the TOC, TN, FA, humin, and CPI were significantly impacted due to synergism among soil properties as positively influenced by the CRP upon conversion of agriculture.

## Introduction

The soil organic carbon (SOC) is the dominant component (~ 58%) of soil organic matter (SOM) and is comprised of a complex mixture of diverse carbonaceous constituents [1, 2]. These constituents include living plant roots and microbes and their metabolites, plant, and animal detritus in various stages of decomposition, and highly decomposed materials that resist further decomposition [3]. The temporal balance among the natural processes of primary production, decomposition, and transformation contribute to SOC formation, partition, and accumulation in terrestrial ecosystems [2, 3]. The SOC is considered as the composite core indicator of soil quality supporting bio-efficiency and diversity, regulating chemical balancing, buffering nutrient cycling, and enhancing ecosystem resilience and services [24].

Temporal land-use change has reportedly impacted SOC accumulation and its lability in natural and human-induced ecosystems [4–6]. Several long-term studies have reported a net loss of SOC upon deforestation or when perennial grasslands are converted into conventionally-tilled (CT) monocropping, except where grasslands are established for previously degraded lands [7, 8]. In contrast, the conversion of routinely plowed cropland, fallow and marginal lands, and degraded forests into pasture or grassland, agriculture, wetlands, or undisturbed conservation zones such as the CRP, has been shown to increase SOC sequestration [9, 10]. Other studies have reported that replacing CT or mono-cropping systems with continuous no-till (NT) and adopting cropping diversity or perennials increased SOC content and lability [6, 7, 11]. It is also reported that marginal cropland conversion to perennial energy crops significantly improved microbiome community, SOC, TN, and other soil quality properties [12, 13].

The SOC contains diverse and multiple C pools which have variable turnover rates [1, 2]. The relative proportions of these pools in any given soil are associated with the amount and quality of inputs and various degrees of temporal changes under contrasting land-use systems [14, 15]. Components of SOC with faster turnover rates (labile pool), often referred to as the active carbon (AC) pool, consist of readily decomposable C compounds such as sugars, amino sugars, amino acids, particulate organic carbon, fulvic acid, as well as root exudates and microbial metabolites [2, 15]. The passive pool, in contrast, has a slower turnover and consists of more recalcitrant C compounds like lignin and humin [1].

Based on classical extraction procedures and the solubility in water as a function of pH, the SOC is conceptually made up of humic acid (HA), fulvic acid (FA), humin, and glucose equivalent total non-humic carbon (NH) pools [1, 16, 17]. While the functionality of the HA, FA, and humin pools is based on its aliphatic and aromatic chemical composition, the NH pool, in contrast, is composed mainly of labile polysaccharides and protein-based compounds [1, 18, 19]. In response to continual microbially-mediated humification processes occurring in SOC formation, labile C pools are often metabolized and partitioned into moderately labile to passive C pools and/or vice-versa [2, 20]. More specifically, these C compounds originate during and/or after plant and animal residue decomposition and subsequent polymerization of metabolites into complex and heterogenic mixture of SOC pools with distinct functional groups, chemical composition, and physico-chemical properties [1, 8]. It is reported that the HA, FA, and humin pools are themselves not biologically passive but rather they accumulate

in soil due to the various degrees of complexations with polyvalent cations and clays with limited microbial access to carry-out ecological functions [17, 21].

While an absolute change in the bulk SOC content takes several years to detect, the AC, a relatively small fraction of labile SOC pool which acts as a source of substrate and energy for microbes can provide early indications of management-induced changes in SOC dynamics [15, 22, 23]. More specifically, the AC significantly influences biological, chemical and physical indicators of soil quality and has been shown to be more sensitive to soil management practices and ecological disturbances than the bulk SOC [2, 23]. Furthermore, maintaining a steady level of labile C such as AC than that of the absolute bulk SOC level is far more important to support biodiversity and efficiency, recycle nutrients, and improve structural stability and hydrological properties [2, 23]. Several studies [2, 23, 24] demonstrated that changes in AC pool, as measured by mild permanganate oxidation, can be predictive of changes in soil functional processes such as N mineralization and aggregate stabilization.

There is an increasing interest in SOC sequestration as a means to offset greenhouse gas emissions and climate change effects [25]. A better understanding of SOC pools and their lability, and how these pools are impacted by land-use change, may facilitate evaluation of a soil's capacity as a sink for, or source of tropospheric $CO_2$, and provide useful information on the land management implications for climate change mitigation and adaptation [26, 27].

Accelerated deforestation is a drastic land-use change that causes rapid decreases in SOC and TN stocks, accelerated emissions of $CO_2$ into the troposphere, degrades soil structural stability, and decreases soil provision of ecosystem services [9, 28]. Land-use change subsequent to deforestation can also impact SOC dynamics [29, 30]. There is a need for more information to evaluate the impacts of drastic land-use change such as deforestation on the depth distribution, stocks, losses, and quality of SOC and TN pools. Further information is also needed on the impacts of land-use practices that occur subsequent to deforestation. While information of land use on SOC dynamics is somewhat available for tropical climates [29, 31], there is not much in the literature for temperate climates.

We hypothesized that in a temperate climate, temporal land-use change including deforestation and subsequent land-use practices would impact variability in site conditions and consequently, impact the SOC and TN and their storage, stratification, quality, and lability. The SOC and TN pools would be affected by the type of vegetation, placement of residues, amounts and quality of inputs, and frequency of disturbances over time. The objectives of the study were to (1) determine the depth distribution of SOC, TN, AC, HA, FA, humin, and NH pools; (2) measure whether depth distribution affected stratification of SOC and TN pools; and (3) evaluate if differences translated into SOC lability and quality upon conversion of forest into agriculture, wetland, the CRP, and bioenergy systems under otherwise similar soil and climatic conditions.

## Materials and methods

### Experimental site and land-use management

The study was conducted on the existing remnant of protected natural forest and its adjacent diverse and temporally managed ecosystems at The Ohio State University South Centers in Piketon (39.0678° N, 83.0092° W), Ohio, USA (**Fig 1**). The original vegetation was primary deciduous forest dominated by oak (*Quercus* spp.), yellow poplar (*Liriodendron tulipifera*), maple (*Acer saccharum*), and hickory (*Annamocarya sinensis*). Around 1900, a large part of the primary forest was converted into intensively managed pasture ecosystems for raising cattle. In the 1950's, the pasture lands were converted into conventionally-tilled agriculture for growing corn and soybeans [32]. In 1991, a portion of the agricultural land was placed under

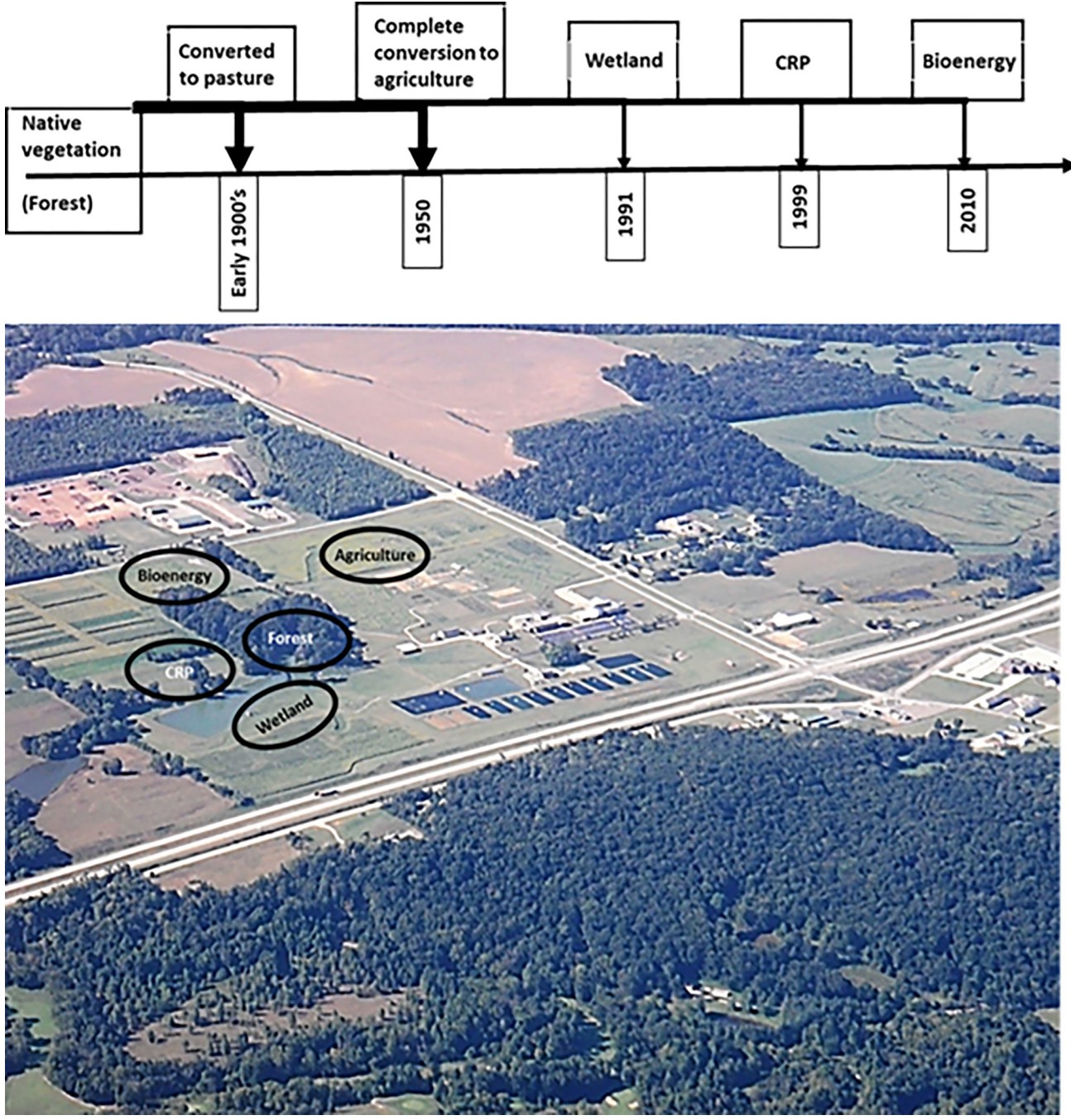

**Fig 1. Chronology of the temporal land use diversity of the study area with sampling locations.**

wetland research study while another portion of the land was placed into the CRP in 1999 with the planting of warm-season prairie grasses, a mixture of switchgrass (*Panicum virgatum* L.), little bluestem (*Schizachyrium scoparium*), big bluestem (*Andropogon gerardii* Vitman), Indian grass (*Sorghastrum nutans*), and Eastern Gamma grass (*Tripsacum dactyloides*). In 2010, a portion of the remaining land was converted into perennial bioenergy plantations with *Miscanthus giganteus* (**Fig 1**).

The monthly air temperatures recorded daily by weather station at the site ranges between 0 to 24°C, relative humidity ranges between 79 to 93%, soil temperatures at 15 cm deep ranges between 3 to 30°C, and solar radiation ranges between 9,980 to 43,040 KW/m². Mean annual rainfall is 96±20 cm, with about 55% of the precipitation falling during the crop growing season (May to September). July is the hottest month and January is the coldest.

## Soil sampling and processing

The soil is a moderately well-drained Omulga silt loam (fine-silty, mixed, mesic *Aeric Fragiaqualf*) that formed in loess, colluvium, and old alluvium with a fragipan below 150 cm depth [32]. Averaged across the 0 to 30 cm depth, the soil contained 30, 15, and 55% of sand, silt, and clay, respectively. Among the land-uses, soil pH varied from 5.7 to 6.3.

Composite soil samples were collected from 0 to 30-cm depth at 7.5 cm intervals. Samples were collected from existing geo-referenced replicated sites (n = 4) in each of the Forest, Agriculture, Bioenergy, CRP, and Wetland experiments. The collected field-moist soils were air-dried under shade at room temperature (~ 25°C) for a period of 15 days, then ground with a porcelain mortar and pestle, and passing through a 2-mm sieve prior to analysis.

## Soil total organic carbon, total nitrogen, and active carbon

The SOC and TN were determined on finely ground (< 125 μm) air-dried soil using the FlashEA-1112 series CNHS-O dry combustion analyzer. The AC, as a measure of labile C, was determined based on $KMnO_4$ oxidation of air-dried soil [2]. A sample of 5 g air-dried soil was taken into a dry 50 ml screw-top polypropylene tube and shaken with 20 ml of slightly acid 0.02M $KMnO_4$ (pH 6) at 100 rpm for 2 min using a reciprocal shaker. The soil-$KMnO_4$ suspension was centrifuged at 2,000 rpm for 20 min to obtain soil-free aliquots and the AC was determined colorimetrically at 550 nm.

## Soil humic acid, fulvic acid, and non-humic carbon

The extraction, separation, and processing of HA and FA were carried out by following the standard procedure of the International Society of Humic Substances [33]. A 5 mL sample of diluted HA or FA aliquot was analyzed for dissolved organic C, using the Shimadzu® Dissolved TOC/TN analyzer. The NH was determined as glucose equivalent C in both HA and FA aliquots by following the anthrone-sulfuric acid method [34]. Humin was calculated after subtracting the total extracted C (FA + HA + NH) from the SOC content.

## Stocks of soil carbon and nitrogen pools

The SOC, AC, TN, HA, FA, humin, and NH stocks (Mg/ha) were calculated by multiplying their respective concentrations with the concurrently measured bulk density (ρb) values at each depth interval (d) and pooled together to calculate for soil profile-wise distribution [7]. The stocks in forested soils were used as SOC and TN loss parameters (control) to calculate the differences in SOC and TN balance under diverse land-use changes upon deforestation.

$$Carbon\ or\ Nitrogen\ stock(Mg\,ha^{-1}) = (ρb \times d \times C\ or\ N \times 10,000)$$

where ρb is the bulk density of the soil (g cm⁻³); C is the soil organic carbon content (g k⁻¹); N is the total nitrogen content (g k⁻¹); and *d* is the soil thickness (cm).

## Stratification of soil carbon and nitrogen pools

Stratification of SOC and TN pools was calculated by dividing their values at different depths under various land-use changes (accumulation or depletion ratios) with the respective values of that SOC and TN in soils under Forest at 22.5 to 30 cm depth [35]. The use of Forest soil lower depth (as a control) to calculate for SOC and TN stratification (accumulation or depletion) is important, so that the impact of deforestation and subsequent temporal land-use changes under similar soil and climate conditions can be compared and contrasted.

## Soil organic carbon and nitrogen accumulation and lability

Using the data on SOC, AC, FA, HA, and NH contents, the C management index (CMI) was calculated [2, 36] as follows:

$$CMI = [CPI \times CLi]$$

where CPI and CLi are the C pool index and C lability index, respectively.

$$CPI = \left[\frac{SOC \; treatment \; soil}{SOC \; in \; the \; control \; soil}\right]$$

$$CLi = \left[\frac{CL \; in \; treatment \; soil}{CL \; in \; the \; control \; soil}\right]$$

where CL refers to the lability of C, which was calculated as follows:

$$CL = \left[\frac{Labile \; C}{Non - labile \; C}\right]$$

Using the same principle of CPI, the NPI was calculated. The labile C pool was considered as the portion of SOC that was measured as the AC, FA, HA, and NH pools, respectively. The non-labile C pool was calculated by subtracting the values of AC, FA, HA, and NH, respectively from the SOC values.

## Soil organic carbon humification coefficients

Several humification quotients, as qualitative measures of SOC, such as humification ratio (HR), humification index (HI), and degree of humification (DH) were calculated [11, 37] as follows:

$$HR(\%) = \left[\left(\frac{HA + FA}{SOC}\right)\right] \times 100,$$

$$HI(\%) = \left[\left(\frac{NH}{HA + FA}\right)\right] \times 100,$$

$$DH(\%) = \left[\left(\frac{HA + FA}{TEC}\right)\right] \times 100.$$

Qualitative variations in SOC were evaluated by spectral characterization of both HA and FA measured at 465 nm ($E_4$) and 665 ($E_6$) nm, respectively using a spectrophotometer [1]. Moreover, the variable $\Delta$log $K$ was calculated as a measure of the degree of humification

(conjugation intensity in FA and HA molecules) following [8].

$$\Delta \log K = \log(\frac{A_{400}}{A_{600}}).$$

Where, the $A_{400}$ and $A_{600}$ were the absorbance values of HA and FA aliquots measured at 400 and 600 nm, respectively in 0.1 M NaOH solution.

## General soil properties

Soil pH (1:2 soil-distilled deionized water suspension) was determined by the glass electrode method. Electrical conductivity in 1:1 soil-water suspension was measured by the electrical conductivity meter. Soil $\rho b$ was determined by following the standard core method. Soil particle size analysis was performed using the Bouyoucos hydrometer method after removing the SOM by $H_2O_2$ oxidation followed by a treatment with 10% $Na_6(PO_3)_6$ solution. The abbreviations and definitions of all the terms used were presented in Table 1.

## Statistical analysis

Significant differences in the concentration, stocks, and stratification of SOC and TN pools and their lability characteristics attributed to the impact of land-use changes following converting primary natural forest were evaluated by a 2-way analysis of variance procedure of the SAS® [38]. Both land-use and soil depth were considered as fixed effects. Simple and interactive effects of predictor variables on dependent variables were separated by the Least Significant Difference (LSD) test with a value of $p \leq 0.05$, unless otherwise mentioned. Both principal components analysis (PCA) and redundancy analysis (RDA) were performed to quantify,

**Table 1. List of abbreviations and definitions.**

| |
|---|
| AC = Active carbon is the labile pool which measured by permanganate oxidation. |
| CL = Carbon lability (labile organic carbon / non-labile organic carbon). |
| CLI = Carbon lability index (CL in treatment soil / CL in control soil). |
| CMI = Carbon management index (CPI x CLI). |
| CPI = Carbon pool index (SOC in treated soil / SOC in control soil) |
| CT = Conventional tillage. |
| DH = Degree of humification [(humic acid + fulvic acid) / total extracted carbon]. |
| $E_4$: $E_6$ = Absorption of fulvic- and humic acid measured at 465 and 665 nm, respectively. |
| FA = Fulvic acid, soluble in both acid and alkaline solutions. |
| $FA_{NH}$ = Fulvic acid associated non-humic carbon. |
| HA = Humic acid, soluble in alkaline solution. |
| $HA_{NH}$ = Humic acid associated non-humic carbon. |
| HI = Humification index [(non-humic carbon / (humic acid carbon + fulvic acid carbon)]. |
| HR = Humification ratio [(humic acid carbon + fulvic acid carbon) / SOC]. |
| NH = Glucose equivalent non-humic carbon. |
| NPI = Nitrogen pool index (N in treatment soil/N in control soil) |
| NT = No-till. |
| $\rho b$ = Soil bulk density. |
| SOM = Soil organic matter. |
| TEC = Total extracted organic carbon by 0.1 M NaOH. |
| TN = Total nitrogen. |
| $\Delta \log K$ = A measure of the degree of humification. |
| $qCO_2$ = Soil microbial cell specific maintenance respiration. |

**Table 2. Impact of deforestation and subsequent land-use diversity on total organic carbon (SOC), total nitrogen (TN), active carbon (AC), fulvic acid (FA), humic acid (HA), humin, and glucose equivalent total non-humic carbon (NH), fulvic acid associated glucose equivalent non-humic carbon ($FA_{NH}$), and humic acid associated glucose equivalent non-humic carbon ($HA_{NH}$) contents at different soil depths.**

| Land-use | Time | Depth | SOC | TN | Humin | AC | FA | HA | NH | $FA_{NH}$ | $HA_{NH}$ |
|---|---|---|---|---|---|---|---|---|---|---|---|
| change | (year) | (cm) | | | (g/kg) | | | | | (mg/kg) | |
| Forest | ----- | 0–30 | 12.4a$^{\neq}$ | 1.1ab | 8.9a | 492.6b | 1.5a | 1.3a | 718a | 388a | 331a |
| Agriculture | 1950 | 0–30 | 8.2c | 1.05b | 5.7b | 506.6ab | 0.7c | 1.3a | 454c | 245c | 209c |
| Wetland | 1991 | 0–30 | 11.2ab | 0.86c | 8.5a | 496.8b | 1.1b | 1.1ab | 434c | 230c | 203c |
| CRP | 1999 | 0–30 | 12.6a | 1.22a | 9.2a | 550.0a | 1.8a | 1.1ab | 600b | 328b | 273b |
| Bioenergy | 2010 | 0–30 | 10.7b | 0.97bc | 8.4a | 519.8a | 0.9c | 1.0b | 599b | 322b | 277b |
| **Land-use x soil depth** | | | | | | | | | | | |
| Forest | ----- | 0–7.5 | 30.0 | 2.38 | 22.3 | 822.9 | 3.1 | 3.1 | 1383 | 849 | 534 |
| | | 7.5–15 | 9.5 | 0.92 | 6.4 | 433.4 | 1.6 | 0.9 | 684 | 300 | 383 |
| | | 15–22.5 | 5.7 | 0.60 | 3.7 | 373.4 | 0.8 | 0.7 | 547 | 287 | 260 |
| | | 22.5–30 | 4.3 | 0.50 | 3.2 | 340.6 | 0.4 | 0.4 | 258 | 114 | 144 |
| Agriculture | 1950 | 0–7.5 | 13.3 | 1.54 | 9.3 | 712.7 | 1.2 | 2.0 | 726 | 380 | 347 |
| | | 7.5–15 | 9.3 | 1.31 | 6.8 | 611.5 | 0.7 | 1.3 | 490 | 240 | 250 |
| | | 15–22.5 | 7.1 | 0.90 | 5.1 | 437.6 | 0.7 | 1.0 | 416 | 262 | 154 |
| | | 22.5–30 | 2.9 | 0.44 | 1.6 | 264.5 | 0.3 | 0.7 | 182 | 97 | 85 |
| Wetland | 1991 | 0–7.5 | 27.4 | 1.28 | 22.4 | 660.3 | 2.2 | 2.1 | 689 | 382 | 307 |
| | | 7.5–15 | 8.5 | 0.96 | 6.0 | 561.9 | 1.0 | 1.0 | 514 | 271 | 243 |
| | | 15–22.5 | 4.6 | 0.77 | 2.9 | 453.0 | 0.7 | 0.6 | 327 | 171 | 156 |
| | | 22.5–30 | 4.2 | 0.42 | 2.8 | 312.0 | 0.6 | 0.6 | 204 | 98 | 106 |
| CRP | 1999 | 0–7.5 | 25.9 | 2.34 | 19.5 | 840.4 | 3.5 | 1.8 | 1041 | 578 | 463 |
| | | 7.5–15 | 13.4 | 1.23 | 9.6 | 611.5 | 2.1 | 1.0 | 667 | 355 | 312 |
| | | 15–22.5 | 6.9 | 0.72 | 4.7 | 431.4 | 1.0 | 0.7 | 446 | 229 | 216 |
| | | 22.5–30 | 4.3 | 0.58 | 3.0 | 316.9 | 0.4 | 0.7 | 247 | 148 | 99 |
| $LSD_{p<0.05}$ | | | | | | | | | | | |
| | Soil depth | | 2.3 | 0.27 | 2.1 | 61.1 | 0.4 | 0.3 | 99.0 | 60.0 | 89 |
| | Land-use x depth | | 4.1 | 0.53 | 3.9 | 136.1 | 0.9 | 0.6 | 200.2 | 119.8 | 187 |

$^{\neq}$ Means separated by same lower-case letter in each column were not significantly different among the land use diversity at $p \leq 0.05$.

contrast, and validate the effects of land use management on SOC and TN pools and associated properties [TOC, TN, humin, AC, FA, HA, NH, $FA_{NH}$, $HA_{NH}$, CPI, and ρb].

## Results and discussion

### Distribution of soil organic carbon and nitrogen pools

Among the land use change, agriculture had the lowest SOC content (8.2 g/kg) when compared to the forest (control), while the CRP had the highest content of SOC (12.6 g/kg) which was similar to that of the forest (**Table 2**). In contrast, the TN content was highest under the CRP (1.22 g/kg) followed by agriculture and lowest under the wetland (0.86 g/kg) when compared to the Forest. As expected, the AC content increased significantly under both CRP and bioenergy compared to the forest and wetland. Like SOC, the humin content was significantly lower under agriculture when compared with other land-use changes. While a higher content of FA was observed in the CRP and forest, a reverse trend was observed in bioenergy and agriculture, respectively. In contrast, similar HA values were measured among the land-use systems. Total NH, FA-bound NH ($FA_{NH}$), and HA-bound NH ($HA_{NH}$) contents were

**Table 3. Impact of deforestation and subsequent land-use diversity on blk density (ρb), total organic carbon (SOC), total nitrogen (TN), active carbon (AC), fulvic acid (FA), humic acid (HA), humin, and glucose equivalent total non-humic carbon (NH), fluvic acid associated glucose equivalent total non-humic carbon (FA_{NH}), and humic acid glucose equivalent total non-humic carbon (HA_{NH}) stocks at different soil depths.**

| Land-use | Time | Depth | Pb | SOC | TN | Humin | AC | FA | HA | NH | FA_{NH} | HA_{NH} |
|---|---|---|---|---|---|---|---|---|---|---|---|---|
| change | (year) | (cm) | (g/cm$^3$) | (Mg/ha) | | | (kg/ha) | | | | | |
| Forest | ------ | 0–30 | 1.20c | 10.9b$^{\neq}$ | 0.97b | 7.9b | 439.7c | 1.3b | 1.2a | 647.7a | 349.7a | 298.0a |
| Agriculture | 1950 | 0–30 | 1.28b | 8.3c | 1.07ab | 5.8c | 517.4b | 0.7c | 1.1a | 617.7ab | 332.1a | 285.5a |
| Wetland | 1991 | 0–30 | 1.23c | 12.1a | 0.92b | 9.3a | 532.6ab | 0.8c | 1.4a | 488.0c | 263.3b | 224.8b |
| CRP | 1999 | 0–30 | 1.23c | 12.2a | 1.19a | 8.9ab | 546.0ab | 1.8a | 1.1a | 612.8b | 334.3a | 278.4a |
| Bioenergy | 2010 | 0–30 | 1.31a | 11.9ab | 0.99b | 9.4a | 570.2a | 1.6a | 1.2a | 491.2c | 260.9b | 230.3b |
| **Land-use x soil depth** | | | | | | | | | | | | |
| Forest | ------ | 0–7.5 | 1.15 | 25.8 | 2.04 | 19.2 | 706.6 | 2.7 | 2.7 | 1189.6 | 730.5 | 459.1 |
| | | 7.5–15 | 1.17 | 8.7 | 0.85 | 5.9 | 398.2 | 1.4 | 0.8 | 627.4 | 275.7 | 351.7 |
| | | 15–22.5 | 1.22 | 5.4 | 0.57 | 3.5 | 355.3 | 0.7 | 0.7 | 522.4 | 273.8 | 248.7 |
| | | 22.5–30 | 1.27 | 3.8 | 0.43 | 2.8 | 298.8 | 0.3 | 0.4 | 226.2 | 99.9 | 126.3 |
| Agriculture | 1950 | 0–7.5 | 1.22 | 13.2 | 1.51 | 9.2 | 703.7 | 1.3 | 1.8 | 927.2 | 526.3 | 400.8 |
| | | 7.5–15 | 1.28 | 9.6 | 1.35 | 7.1 | 630.1 | 0.8 | 0.9 | 728.1 | 377.6 | 350.4 |
| | | 15–22.5 | 1.30 | 7.5 | 0.94 | 5.3 | 456.3 | 0.4 | 0.7 | 477.1 | 249.1 | 228.0 |
| | | 22.5–30 | 1.31 | 3.0 | 0.47 | 1.7 | 279.3 | 0.2 | 0.8 | 314.2 | 161.0 | 153.2 |
| Wetland | 1991 | 0–7.5 | 1.17 | 30.3 | 1.41 | 24.8 | 726.7 | 1.3 | 2.3 | 802.7 | 419.6 | 383.1 |
| | | 7.5–15 | 1.20 | 8.8 | 0.99 | 6.2 | 579.0 | 0.7 | 1.3 | 505.2 | 247.6 | 257.6 |
| | | 15–22.5 | 1.25 | 4.9 | 0.81 | 3.1 | 479.2 | 0.7 | 1.0 | 442.2 | 278.6 | 163.5 |
| | | 22.5–30 | 1.28 | 4.6 | 0.47 | 3.1 | 345.4 | 0.4 | 0.8 | 200.8 | 106.8 | 94.0 |
| CRP | 1999 | 0–7.5 | 1.19 | 23.0 | 2.07 | 17.3 | 743.8 | 3.1 | 1.6 | 924.2 | 513.3 | 410.9 |
| | | 7.5–15 | 1.22 | 13.8 | 1.27 | 9.9 | 631.7 | 2.2 | 1.0 | 689.1 | 366.5 | 322.6 |
| | | 15–22.5 | 1.23 | 7.3 | 0.77 | 5.0 | 459.5 | 1.1 | 0.8 | 475.8 | 244.7 | 231.0 |
| | | 22.5–30 | 1.27 | 4.7 | 0.65 | 3.3 | 349.0 | 0.4 | 0.8 | 270.5 | 161.7 | 108.9 |
| Bioenergy | 2010 | 0–7.5 | 1.25 | 16.4 | 2.13 | 12.6 | 741.4 | 2.1 | 1.9 | 645.6 | 357.9 | 287.7 |
| | | 7.5–15 | 1.26 | 17.1 | 0.82 | 14.2 | 626.5 | 2.0 | 1.2 | 640.5 | 337.4 | 303.0 |
| | | 15–22.5 | 1.31 | 10.6 | 0.49 | 8.7 | 472.2 | 1.4 | 0.8 | 404.6 | 211.0 | 193.6 |
| | | 22.5–30 | 1.38 | 3.5 | 0.52 | 2.1 | 440.7 | 0.9 | 0.6 | 226.7 | 108.5 | 118.2 |
| LSD$_{p<0.05}$ | | | | | | | | | | | | |
| | | Soil depth | 0.06 | 0.5 | 0.22 | 0.4 | 60.4 | 0.4 | 0.3 | 202.7 | 158.3 | 93.1 |
| | | Land-use x depth | 0.13 | 1.1 | 0.45 | 0.9 | 121 | 0.7 | 0.6 | ns | ns | ns |

$^{\neq}$ Means separated by same lower-case letter in each column were not significantly different among the land use diversity at p≤0.05.

significantly higher in the forest, intermediate in bioenergy and the CRP, and lowest in wetland and agriculture, respectively.

When converting the concentration of SOC and TN pools into mass per unit of area (stock), a significant but slightly different impact of land use on SOC and TN stocks was observed (**Tables 2 and 3**). Likewise, when the SOC, TN, AC, FA, HA, humin, NH, FA_{NH}, and HA_{NH} stocks at each depth interval were pooled, their profile-wise (0 to 30 cm) distribution varied significantly by the impact of land-use change (**Table 4; Fig 2**).

While the SOC stock was higher (by 43 to 47%) under bioenergy, wetland, and the CRP than under agriculture, it was only 9 to 12% higher when compared to forest (**Fig 2**). Agriculture, over the years, had significantly decreased the SOC stocks by 31% upon deforestation but increased TN by 9% when compared to forest. The AC stock did increase (by 18 to 30%) under all land uses when compared with the forest and the highest increase was observed

**Table 4. Impact of deforestation and subsequent land-use diversity on total organic carbon (SOC), total nitrogen (TN), active carbon (AC), fulvic acid (FA), humic acid (HA), humin, and glucose equivalent total non-humic carbon (NH), fluvic acid associated glucose equivalent total non-humic carbon (FA$_{NH}$), and humic acid glucose equivalent total non-humic carbon ($_{HANH}$) stocks in soil profile (0–30 cm).**

| Land-use | Time | SOC | TN | Humin | AC | FA | HA | NH | FA$_{NH}$ | HA$_{NH}$ |
|---|---|---|---|---|---|---|---|---|---|---|
| change | (year) | | | | | (Mg/ha) | | | | |
| Forest | ----- | 43.7b$^{\neq}$ | 3.9c | 31.2c | 1.76b | 5.3b | 4.6a | 2.6a | 1.4a | 1.2a |
| Agriculture | 1950 | 33.3c | 4.27b | 23.8d | 2.07a | 2.7c | 4.3a | 2.5a | 1.3ab | 1.2a |
| Wetland | 1991 | 48.5a | 3.69d | 38.0a | 2.13a | 3.1c | 5.4a | 2.0b | 1.1b | 0.9b |
| CRP | 1999 | 48.9a | 4.76a | 34.8b | 2.18a | 7.2a | 4.4a | 2.5a | 1.3ab | 1.2a |
| Bioenergy | 2010 | 47.6a | 3.95c | 35.7b | 2.28a | 6.4a | 4.8a | 2.0b | 1.1b | 0.9b |

$^{\neq}$ Means separated by same lower-case letter in each column were not significantly different among the land use diversity at p≤0.05.

under bioenergy. The FA stock, in contrast, consistently decreased under agriculture and wetland but increased under the CRP and bioenergy compared to forest. The NH, FA$_{NH}$, and HA$_{NH}$ stocks under forest were statistically similar with agriculture, and the CRP, but significantly lower under wetland.

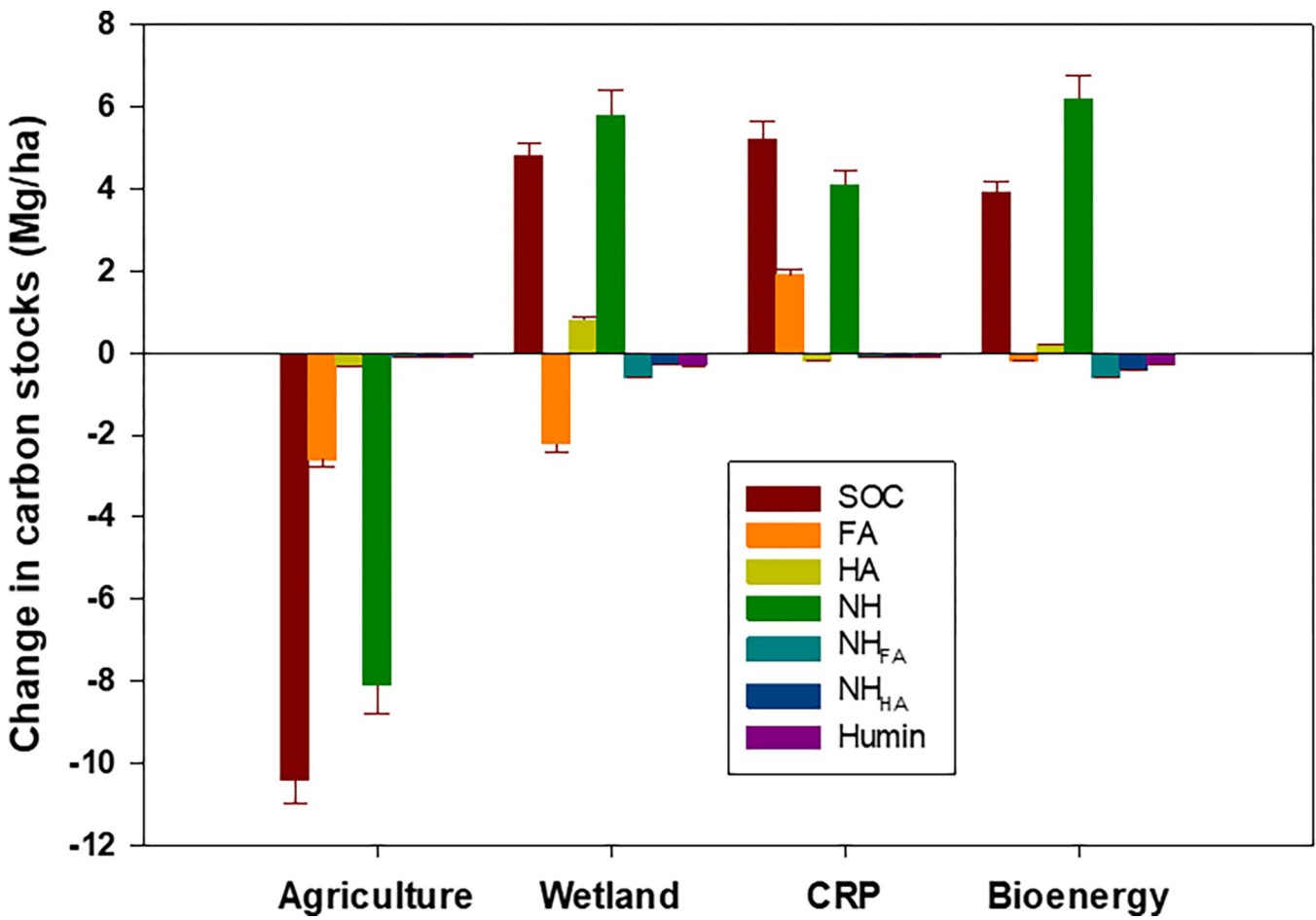

**Fig 2. Impact of deforestation and subsequent land-use diversity on change in total organic carbon (SOC), total nitrogen (TN), active carbon (AC), fulvic acid (FA), humic acid (HA), humin, and glucose equivalent total non-humic carbon (NH), fluvic acid associated glucose equivalent total non-humic carbon (FA$_{NH}$), and humic acid glucose equivalent total non-humic carbon (HA$_{NH}$) stocks in soil profile (0–30 cm), relative to the forest.** Data were presented with mean ± standard error.

While the SOC, TN, AC, FA, HA, NH, $FA_{NH}$, $HA_{NH}$, and humin contents significantly decreased by depth with a land-use × depth, in contrast, the $\rho b$ values increased by depth with a land use x depth. The SOC, TN, humin, AC, FA, HA, NH, $FA_{NH}$, and $HA_{NH}$ stocks were also decreased by depth without a consistent land use x depth.

A significant difference in SOC and TN pools, including the AC, FA, NH, $FA_{NH}$, and $HA_{NH}$ as shown by the impact of land-use changes was expected due to variations in the (i) type and age of the vegetative covers, (ii) quality, amount, and placement of inputs, and (iii) intensity of disturbances [11, 39]. When compared with forest as the control, a significant loss in SOC under agriculture was due to the impact of annual plowing. Plowing promotes nutrient mineralization under warmer and oxic soil conditions and accelerates opportunistic microbial decomposition of fragmented plant residues intimately mixed with the soil, and induces priming effects on native SOC [11, 40]. Soil disturbances by plowing that repeatedly disrupt macroaggregates are likely to shift the balance of interacting biochemical mechanisms toward microbial catabolism, resulting in net loss of SOC [11]. In order to ameliorate such a disturbed soil ecosystem alike agriculture, it is recommended to adopt conservation tillage or no-tillage to enhance aggregate formation [7, 41] to protect the macroaggregate-associated carbon and nitrogen from opportunistic decomposers. It is also recommended to use biochar to accumulate both labile and stable fractions of SOC [7, 42]. Adding perennials, cover crops and the use of green manure crops also improves labile C to support soil ecosystem functions and services [2, 23].

Higher SOC contents under the CRP and forest were primarily due to reduced contact between microbes and surface accumulation of unfragmented litterfalls under undisturbed, moist, cooler, and partially anaerobic site conditions with a dominance of energy efficient fungal food webs [7, 40]. Consequently, the transformation of plant-derived C into SOC pools via humification processes was more biologically efficient under the CRP than under highly disturbed agricultural ecosystem [7, 39, 41]. Similarly, a higher SOC under wetland was associated with slower microbial activity, limited $O_2$ diffusion, and lower water temperatures [42, 43]. A faster recovery of SOC with the establishment of bioenergy plantations was due to the transition from historical long-term plowing agricultural system to no-till bioenergy system. Likewise, a significantly higher TN content under the CRP was related to C: N stoichiometry and the contribution of biological N fixation by leguminous herbs and weeds (*Mimosa* and *Kudzu spp.*). In contrast, a drastic reduction in TN content under wetland was due to gaseous N loss associated with anaerobic microbially regulated processes of denitrification [42–44].

A significantly higher AC under the CRP and bioenergy was due to greater rhizodeposition and fine roots contribution as impacted by extensive and deep-rooted perennial warm season prairie grasses (a mixture of switchgrass, Indian grass, big bluestem, and Eastern gamma grass) and by *Miscanthus* spp. Due to their high biomass production under undisturbed site conditions, these $C_4$ grasses substantially contributed rhizodeposition to the synthesis of AC. As the AC, a small pool of labile SOC, an increase in SOC content under the CRP is expected to a substantial increase in AC content. A greater depletion of FA under agriculture may have resulted from accelerated catabolism (microbial cell maintenance respiration, $qCO_2$) of more labile FA by bacterial food web dominance and edge-of-field loss via leaching and runoff from plowed fields after major rainstorms and early spring snow melts [2, 45]. It is reported that among the humified compounds, the FA was influenced more than that of the HA, humin, and SOC, respectively, by the impact of land-use changes [46]. A relatively low FA but high HA content under agriculture, with respect to forest, indicated an accumulation of higher molecular weight aromatic and stabilized HA and humin compounds, which are likely formed due to complex physico-chemical interactions with clays and polyvalent cations [1]. In contrast, a higher content of FA under forest and the CRP is related to an accumulation of low molecular weight

labile aliphatic C compounds that are perhaps more polar and weakly associated with the cations or clays [1]. Thus, the FA may be a more sensitive and an early indicator than the HA when evaluating the impact of land-use changes of SOC quality.

A partial recovery in NH content under the CRP and bioenergy with respect to forest was attributable to the increased release and translocation of nonstructural carbohydrates from diverse plant residues, extensive fine root production, and greater root exudation and sloughing [47]. These labile C compounds, in turn, are likely metabolized by microbes to release extracellular polysaccharides as binding agents to form macroaggregates from microaggregates and protect as particulate organic matter within aggregates [7, 48]. In contrast, a reduced level of NH under both agriculture and wetland may have resulted from accelerated utilization of labile C by bacteria-dominated opportunistic food webs, leaching and runoff, and biologically mediated gaseous loss of C via aerobic and anaerobic processes [42–44]. The significant land-use x depth interaction suggested that forest, the CRP, and bioenergy systems may have the most favorable biological conditions to process residues into SOC retained and stratified more at the surface than that of the agriculture, where residue was fragmented and incorporated thoroughly into the warm, and aerobic soil ecosystems by the impact of plowing.

## Stratification of soil organic carbon and nitrogen pools

The stratification of SOC and TN pools decreased along the gradient of the historical disturbance intensity, from forest to agriculture. For example, the SOC stratification decreased significantly (by 52%) under agriculture followed by bioenergy and wetland; however, the CRP had a similar stratification pattern when compared with forest (**Table 5**).

The TN and humin stratification patterns observed were similar to that of the SOC. In contrast, the AC stratification was highest under the CRP than that of other land uses. As expected, the FA stratification was consistently lower under both agriculture and wetland, but higher under the CRP, when compared to forest. Both forest and wetland had the highest HA stratification than that of the CRP, bioenergy, and agriculture, respectively. Likewise, the NH, $FA_{NH}$, and $HA_{NH}$ stratification was highest in forest and lowest in wetland and bioenergy. Averaged across land-use systems, the stratification of SOC and TN pools decreased with depth without a consistent land-use x depth interaction except that observed with the AC.

Stratification of SOC and soil nutrients is common in undisturbed ecosystems such as grasslands and forests, as well as when degraded cropland is restored or annually plowed lands placed with continuous NT or perennial vegetation [35, 49]. Contrary to absolute contents, the stratification of SOC pools in our study suggested that the CRP and bioenergy systems were acting as C sinks. Like forest, the perennial Miscanthus and warm season mixed prairie vegetative systems (e.g., the CRP and bioenergy) would be expected to provide greater and diverse quantities of residues to surface soil, which in turn, are expected to increase SOC stratification. The higher stratification of SOC and TN pools reflects a physically undisturbed ecosystem that leads to better soil quality [35]. In contrast, a depletion in SOC stratification results from annual tillage-associated inversion and mixing of surface soil with the subsurface soil in the agriculture ecosystem.

## Characteristics and lability of soil organic carbon and nitrogen pools

Agriculture had the wider HA: FA ratios followed by wetland, compared to forest; however, the CRP, forest, and bioenergy had the similar HA: FA ratios among themselves (**Fig 3**).

The HI (humification index), as one of the indicators of SOC quality, was highest in agriculture (36.5%) and lowest in the CRP and wetland (ranging from 20.9 to 22.3%), when compared to forest. In contrast, both DH (degree of humification) and HR (humification ratio) of SOC

**Table 5. Impact of deforestation and subsequent land-use diversity on the stratification of total organic carbon (SOC), total nitrogen (TN), active carbon (AC), fulvic acid (FA), humic acid (HA), humin, and glucose equivalent total non-humic carbon (NH), fluvic acid associated glucose equivalent total non-humic carbon (FA$_{NH}$), and humic acid glucose equivalent total non-humic carbon (HA$_{NH}$) pools at different soil depths.**

| Land-use | Time | Depth | SOC | TN | Humin | AC | FA | HA | NH | FA$_{NH}$ | HA$_{NH}$ |
|---|---|---|---|---|---|---|---|---|---|---|---|
| Change | (year) | (cm) | | (values were divided by the values at 22.5–30 cm depth of forest) | | | | | | | |
| Forest | ------ | 0–30 | 2.88a$^{\neq}$ | 2.20b | 2.78a | 1.44b | 3.66b | 3.22a | 2.78a | 3.39a | 2.29a |
| Agriculture | 1950 | 0–30 | 1.90c | 2.11b | 1.79b | 1.49b | 1.66d | 2.59b | 2.32a | 2.82b | 1.92b |
| Wetland | 1991 | 0–30 | 2.61b | 1.72d | 2.67a | 1.46b | 1.82d | 3.15a | 1.76b | 2.14c | 1.45c |
| CRP | 1999 | 0–30 | 2.94a | 2.43a | 2.87a | 1.61a | 4.42a | 2.68b | 2.32a | 2.87b | 1.89b |
| Bioenergy | 2010 | 0–30 | 2.50b | 1.93c | 2.64a | 1.52ab | 2.82c | 2.67b | 1.68b | 2.02c | 1.41c |
| **Land-use x soil depth** | | | | | | | | | | | |
| Forest | ------ | 0–7.5 | 6.97 | 4.75 | 6.98 | 2.41 | 7.83 | 7.80 | 5.35 | 7.44 | 3.70 |
| | | 7.5–15 | 2.22 | 1.84 | 2.01 | 1.27 | 3.92 | 2.14 | 2.65 | 2.63 | 2.66 |
| | | 15–22.5 | 1.33 | 1.19 | 1.14 | 1.10 | 1.91 | 1.85 | 2.12 | 2.51 | 1.80 |
| | | 22.5–30 | 1.00 | 0.99 | 1.01 | 1.00 | 0.96 | 1.08 | 1.00 | 1.00 | 1.00 |
| Agriculture | 1950 | 0–7.5 | 3.09 | 3.09 | 2.92 | 2.09 | 3.20 | 4.58 | 3.62 | 4.66 | 2.81 |
| | | 7.5–15 | 2.17 | 2.63 | 2.14 | 1.79 | 1.85 | 2.16 | 2.73 | 3.20 | 2.35 |
| | | 15–22.5 | 1.66 | 1.80 | 1.58 | 1.28 | 0.99 | 1.67 | 1.76 | 2.08 | 1.51 |
| | | 22.5–30 | 0.67 | 0.89 | 0.51 | 0.78 | 0.58 | 1.95 | 1.15 | 1.33 | 1.00 |
| Wetland | 1991 | 0–7.5 | 6.37 | 2.57 | 7.01 | 1.94 | 2.98 | 5.12 | 2.81 | 3.33 | 2.40 |
| | | 7.5–15 | 1.98 | 1.93 | 1.87 | 1.65 | 1.79 | 3.22 | 1.90 | 2.11 | 1.73 |
| | | 15–22.5 | 1.06 | 1.53 | 0.91 | 1.33 | 1.66 | 2.46 | 1.61 | 2.30 | 1.07 |
| | | 22.5–30 | 0.97 | 0.85 | 0.88 | 0.91 | 0.86 | 1.80 | 0.70 | 0.85 | 0.59 |
| CRP | 1999 | 0–7.5 | 6.03 | 4.67 | 6.10 | 2.46 | 8.84 | 4.59 | 4.03 | 5.06 | 3.21 |
| | | 7.5–15 | 3.11 | 2.46 | 2.99 | 1.79 | 5.36 | 2.45 | 2.58 | 3.11 | 2.17 |
| | | 15–22.5 | 1.60 | 1.45 | 1.45 | 1.27 | 2.57 | 1.86 | 1.72 | 2.01 | 1.50 |
| | | 22.5–30 | 1.01 | 1.15 | 0.94 | 0.93 | 0.91 | 1.81 | 0.96 | 1.29 | 0.69 |
| Bioenergy | 2010 | 0–7.5 | 4.06 | 4.68 | 4.19 | 2.34 | 5.50 | 5.17 | 2.66 | 3.34 | 2.13 |
| | | 7.5–15 | 3.19 | 1.33 | 3.57 | 1.49 | 2.56 | 2.48 | 1.99 | 2.37 | 1.69 |
| | | 15–22.5 | 1.99 | 0.79 | 2.20 | 1.12 | 1.79 | 1.57 | 1.26 | 1.49 | 1.08 |
| | | 22.5–30 | 0.74 | 0.93 | 0.58 | 1.15 | 1.43 | 1.46 | 0.79 | 0.85 | 0.74 |
| **LSD$_{p<0.05}$** | | | | | | | | | | | |
| | Soil depth | 0.4 | 0.54 | 0.30 | 0.18 | 0.40 | 0.6 | 0.40 | 0.20 | 0.70 | |
| | Land-use x depth | ns | 1.2 | ns | 0.4 | ns | ns | ns | ns | ns | |

$^{\neq}$ Means separated by same lower-case letter in each column were not significantly different among the treatments at p≤0.05.

were variably impacted by the land-use change. The DH was significantly higher but statistically similar among bioenergy, the CRP, and wetland (82 to 83%) than that of the agriculture and forest (73 to 77%). The HR values were lower in bioenergy (27.3%) when compared to the highest HR values in agriculture, the CPR, forest, and wetland, respectively.

Spectral analyses have shown higher E$_4$: E$_6$ ratios for the FA than that of the HA in all the land-use systems (Fig 4). The E$_4$: E$_6$ ratio of the FA was highest in forest (16.2) followed by the CRP (12.1), bioenergy (11.8), and wetland (11.3), when compared to the lowest (9.7) in agriculture. In contrast, the E$_4$: E$_6$ ratio of the HA was higher in the CRP (5.9) relative to other land-use systems (4.5 to 5.1). Likewise, the Δlog $K$ values of both FA and HA did influence by land-use change. While the wetland had the highest Δlog $K$ values of FA, the agriculture, bioenergy and the CRP had the lowest Δlog $K$ values of FA. In contrast, forest and wetland had the lowest Δlog $K$ values of HA and bioenergy and the CRP had the highest Δlog $K$ values of HA.

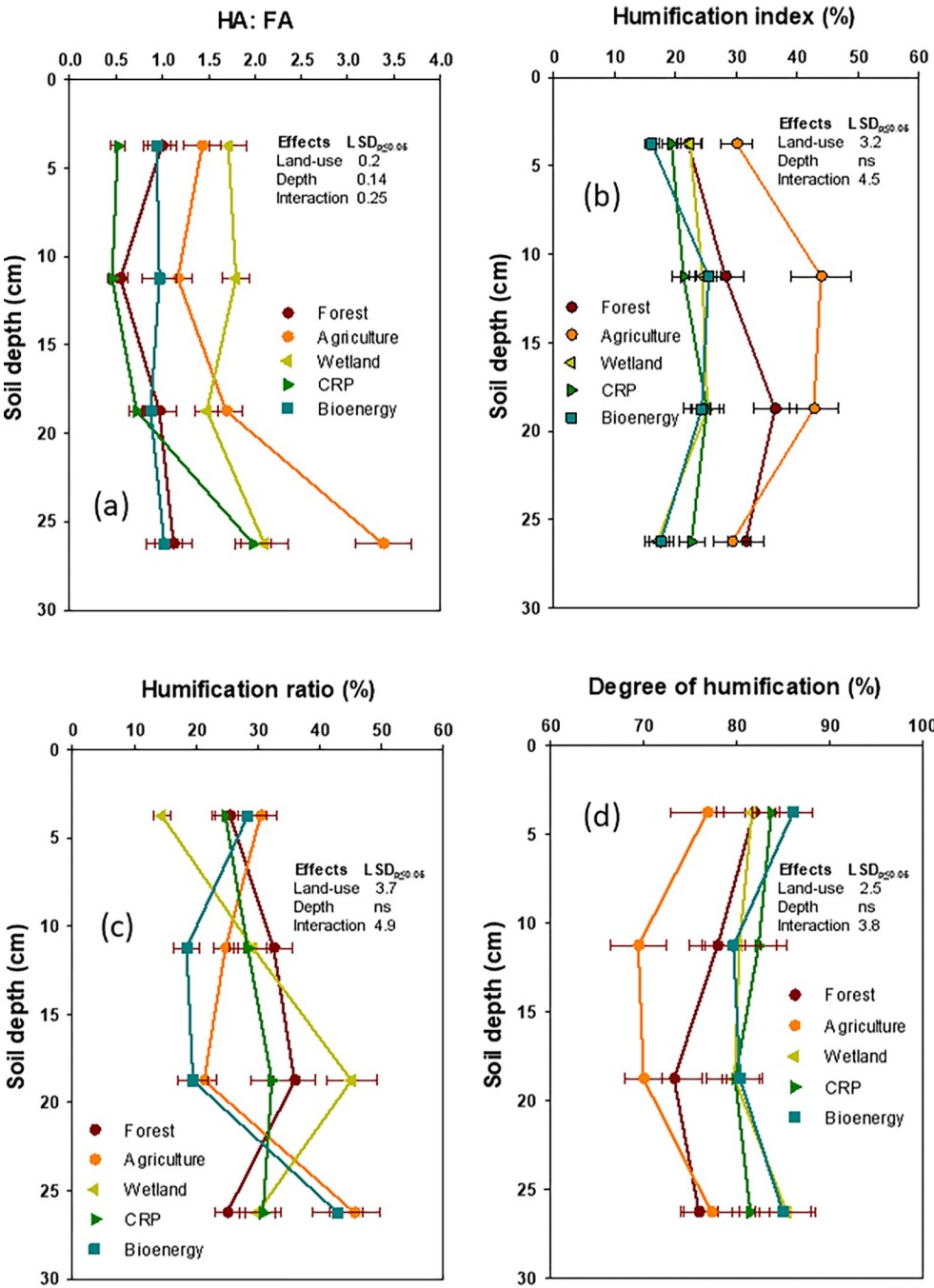

**Fig 3. Impact of deforestation and subsequent land-use diversity on humic acid (HA) and fulvic acid (FA) distribution and humification characteristics of organic carbon at different soil depths (data were presented with mean ± standard error).**

The accumulation or depletion of SOC pools and their lability as evaluated by CPI and CMI variably affected by land-use diversity (**Table 6**; **Figs 5 and 6**). While the CPI decreased by 10 to 16% under wetland and agriculture but it increased by 7 to 12% under bioenergy and the CRP, when compared to forest. A similar response was observed on NPI. The SOC lability (CL) as determined via AC, FA, and HA was highest under wetland and the CRP, but lowest

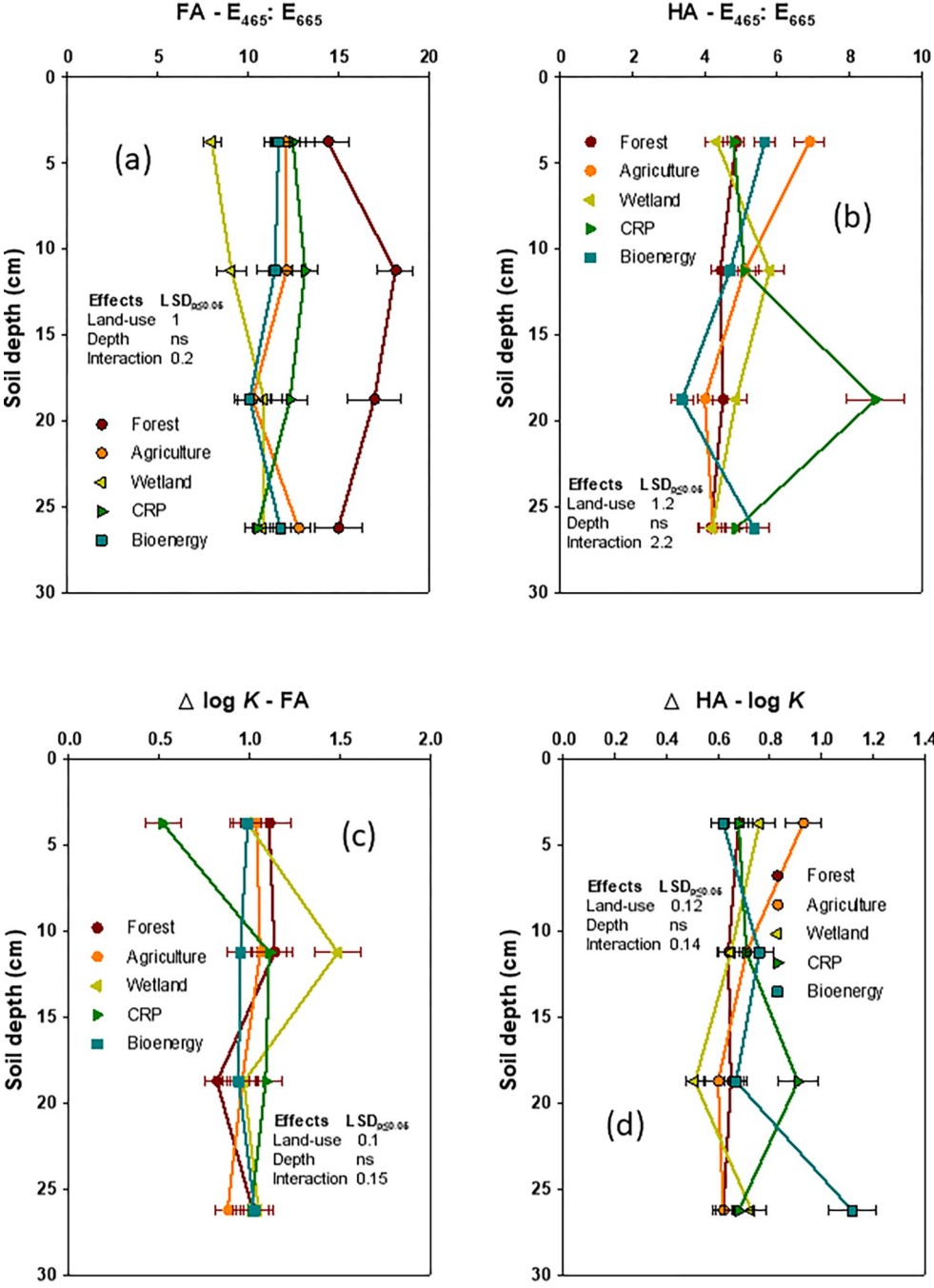

**Fig 4. Impact of deforestation and subsequent land-use diversity on spectral characteristics of fluvic acid (FA) and humic acid (HA) components of total organic carbon at different soil depths (data were presented with mean ± standard error).**

under forest as determined via AC. Lowest values of CL were observed under wetland, the CRP and agriculture when determined via NH (**Fig 5**). A similar response of land use diversity on the depth distribution of the SOC lability indices (CLI) was observed when compared to the forest (**Fig 6**).

**Table 6. Impact of deforestation and subsequent land-use diversity on the carbon and nitrogen pool (CPI and NPI) and management indices (CMI), calculated based on active carbon (AC), fulvic acid (FA), humic acid (HA), and glucose equivalent total non-humic carbon (NH) pools at different soil depths.**

| Land-use | Time | Depth | CPI | NPI | Carbon management index (CMI) | | | |
|---|---|---|---|---|---|---|---|---|
| change | (year) | (cm) | | | AC | FA | HA | NH |
| Forest | ------ | 0–30 | 1.00a≠ | 1.00b | 1.00b | 1.00b | 1.00b | 1.00a |
| Agriculture | 1950 | 0–30 | 0.84b | 0.93bc | 1.1ab | 0.9b | 0.98b | 0.63b |
| Wetland | 1991 | 0–30 | 0.90b | 0.82c | 1.22a | 0.47c | 1.21b | 0.93a |
| CRP | 1997 | 0–30 | 1.12a | 1.17a | 1.18a | 1.21a | 1.12b | 0.86ab |
| Bioenergy | 2010 | 0–30 | 1.07a | 1.12a | 1.11ab | 0.78b | 1.77a | 0.85ab |
| **Land-use x soil depth** | | | | | | | | |
| Forest | ------ | 0–7.5 | 1.00 | 1.00 | 1.00 | 1.00 | 1.00 | 1.00 |
| | | 7.5–15 | 1.00 | 1.00 | 1.00 | 1.00 | 1.00 | 1.00 |
| | | 15–22.5 | 1.00 | 1.00 | 1.00 | 1.00 | 1.00 | 1.00 |
| | | 22.5–30 | 1.00 | 1.00 | 1.00 | 1.00 | 1.00 | 1.00 |
| Agriculture | 1950 | 0–7.5 | 0.44 | 0.54 | 0.86 | 0.33 | 0.31 | 0.24 |
| | | 7.5–15 | 0.98 | 1.05 | 1.34 | 0.68 | 1.31 | 0.82 |
| | | 15–22.5 | 1.25 | 1.28 | 1.28 | 1.50 | 1.34 | 0.91 |
| | | 22.5–30 | 0.67 | 0.85 | 0.93 | 1.07 | 0.97 | 0.54 |
| Wetland | 1991 | 0–7.5 | 0.91 | 0.98 | 1.28 | 0.62 | 0.92 | 1.08 |
| | | 7.5–15 | 0.90 | 0.72 | 1.31 | 0.26 | 0.61 | 0.63 |
| | | 15–22.5 | 0.80 | 0.66 | 1.13 | 0.25 | 0.46 | 0.42 |
| | | 22.5–30 | 0.97 | 0.93 | 1.17 | 0.77 | 2.84 | 1.57 |
| CRP | 1997 | 0–7.5 | 0.86 | 0.98 | 1.05 | 1.17 | 0.57 | 0.75 |
| | | 7.5–15 | 1.41 | 1.34 | 1.44 | 1.37 | 1.13 | 0.96 |
| | | 15–22.5 | 1.20 | 1.21 | 1.19 | 1.37 | 0.98 | 0.79 |
| | | 22.5–30 | 1.01 | 1.15 | 1.03 | 0.94 | 1.80 | 0.95 |
| Bioenergy | 2010 | 0–7.5 | 0.58 | 0.65 | 0.91 | 0.49 | 0.91 | 0.70 |
| | | 7.5–15 | 1.45 | 1.43 | 1.47 | 0.61 | 2.33 | 1.04 |
| | | 15–22.5 | 1.50 | 1.50 | 1.28 | 1.00 | 1.61 | 0.88 |
| | | 22.5–30 | 0.74 | 0.89 | 0.78 | 1.01 | 2.21 | 0.78 |
| LSD$_{p<0.05}$ | Soil depth | | ns | ns | ns | ns | 0.56 | ns |
| | Land-use x depth | | 0.45 | 0.47 | 0.31 | 0.70 | 1.20 | 0.56 |

≠ Means separated by same lower-case letter in each column were not significantly different among the treatments at p≤0.05.

In contrast, the CMI, as a composite measure of SOC accumulation or depletion and lability, calculated based on AC, FA, HA, and NH pools, variably affected by land-use change. The CMI based on AC was higher under wetland and the CRP when compared to forest (**Table 6**). In contrast, the CMI based on FA was lowest under wetland but highest under the CRP than that of other land uses. The CMI based on HA was highest under Bioenergy than that other land uses. When using NH, the CMI in agriculture decreased, when compared to forest.

The HA: FA ratio significantly increased with depth and affected by land-use x depth interaction. Likewise, the $E_4$: $E_6$ ratio of HA and Δlog $K$ values of both FA and HA were influenced by land use x depth. The land-use x depth significantly influenced the CPI, NPI, CL, CLI, and CMI, respectively.

Significant differences in HA: FA ratio were expected due to the variations in vegetative composition and the microbial palatability and/or biodegradability of the residues as influenced by the impact of land-use changes [50]. An accelerated microbial catabolism and edge-of-field loss of soluble FA compared to the stable and insoluble HA, may have resulted in a significant increase in the HA: FA ratio under agriculture relative to other land-use systems. The

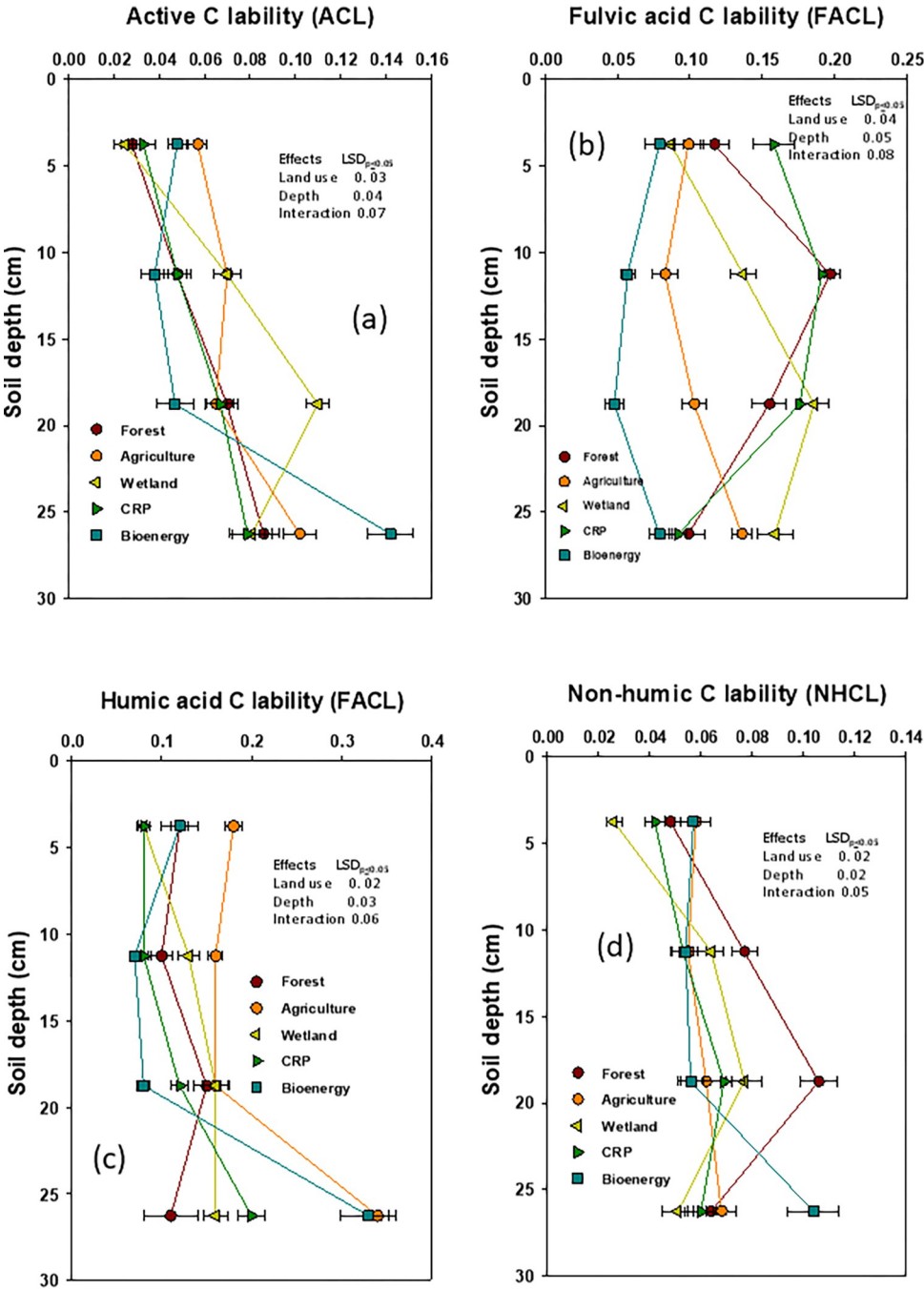

**Fig 5. Impact of deforestation and subsequent land-use diversity on active carbon (AC), fulvic acid (FA), humic acid (HA), and glucose equivalent total non-humic carbon (NH) lability at different soil depths (data were presented with mean ± standard error).**

wider HA: FA ratios under agriculture may also be attributed to the prevalence of higher proportions of high molecular weight polymerized and aromatic C compounds of HA origin [46, 51–53]. In contrast, the narrower HA: FA ratios in the CRP and forest reflect the presence of a relatively large proportion of low molecular weight aliphatic C compounds [1, 51]. The low molecular weight aliphatic C compounds, heavily substituted with -OH (alcoholic) and

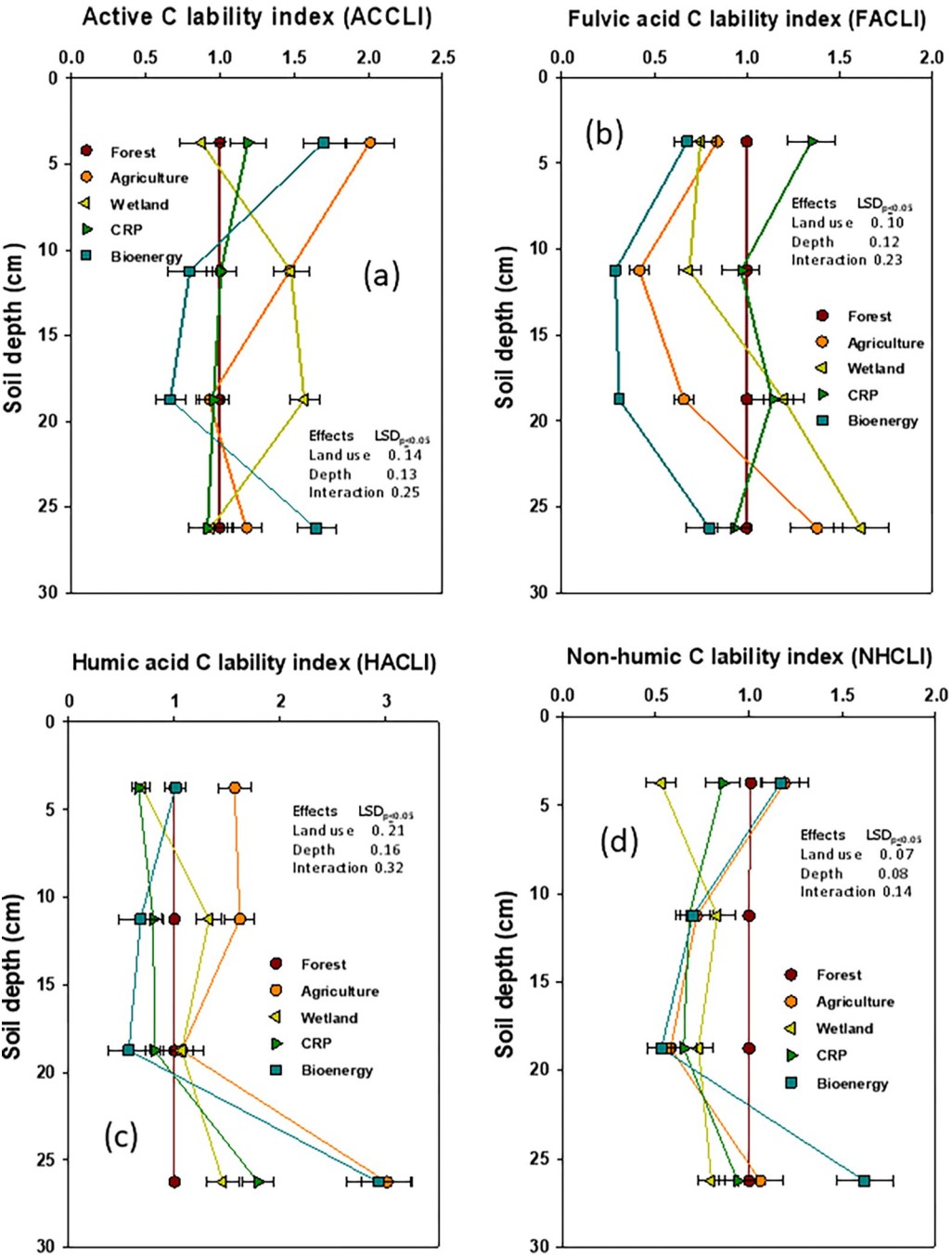

**Fig 6. Impact of deforestation and subsequent land-use diversity on active carbon (AC), fulvic acid (FA), humic acid (HA), and glucose equivalent total non-humic carbon (NH) lability indices at different soil depths (data were presented with mean ± standard error).**

-COOH functional groups, are typical of FA composition, which accumulates in undisturbed and biologically diverse healthy soils [52, 54].

Significantly higher HI and HR values under agriculture relative to other land-use systems suggested a more humified and stable SOC quality [37, 54]. Results of a 40-year study have reported that a narrower HA: FA ratio was associated with a higher DH in SOC under

annually plowed soil when compared with the unplowed grassland soil [37, 54]. A higher DH in SOC under agriculture is due to the presence of a higher proportion of HA released during rapid decomposition of residues by aerobic bacteria-dominated food webs, followed by greater polymerization and aromatic condensation, which is facilitated by the impact of plowing [1, 53]. Hence, lower $\Delta$log $K$ values of HA when compared with the FA indicated that a higher DH is associated with HA under agriculture [8, 55]. Judging from the greater enrichment of the FA in SOC, the CRP and bioenergy ecosystems might have been dominated by energy efficient biochemical pathways that are responsible for reduced aromatic condensation with a concomitant increased in aliphatic nature of SOC.

The wider $E_4$: $E_6$ ratios of FA under the CRP, forest, and bioenergy reflects a SOC with smaller degrees of aromatic condensation and presence of a relatively large proportion of aliphatic structures containing higher total acidity with the dominance of -COOH groups [1, 56]. In contrast, the narrower $E_4$: $E_6$ ratios of HA under agriculture, suggested that SOC had high degrees of aromatic condensation with low total acidity but dominance of phenolic -OH groups [1, 56]. As the $E_4$: $E_6$ ratio is governed by the molecular size, where higher $E_4$: $E_6$ ratios under the CRP are associated with smaller sized HA molecules, whereas lower $E_4$: $E_6$ ratios under agriculture indicates the presence of larger sized HA molecules [21, 51]. Our results closely agree with the results of previous studies [51, 57].

While the higher values of CPI and NPI under the CRP suggested that SOC and TN are accumulating, the smaller CPI and NPI values under agriculture and wetland, in contrast, suggesting a net loss of SOC and TN contents over time. A significant linear relationship ($R^2$ = 0.89) between the CPI and NPI values accounted for 89% of the total variability in the NPI and vice-versa to justify the C: N stoichiometry in SOM in response to land-use change (**Fig 7**).

However, the slope ($\Delta$NPI: $\Delta$CPI = 0.88) of the relationship suggested that proportionally a slightly more SOC was sequestering when compared to the TN accumulation or depletion in response to land-use change. Likewise, the higher CMI values under the CRP indicated the positive impact of the CRP management on SOC lability. While the CMI with higher values under the CRP suggested an accumulation of labile pool of SOC, in contrast, the smaller values indicating a depletion of labile pool of SOC under agriculture [7, 36].

Based on chemical, spectroscopic, and lability analyses on SOC, our results suggested that the CRP may have favored an accumulation of SOC with higher proportions of labile and aliphatic C compounds, whereas Agriculture may have favored an accumulation of SOC with high proportions of non-labile and aromatic C compounds.

## Principal components and redundancy analyses

The PCA performed for all treatments and based on all data associated with measured soil variables clearly separated and/or discriminated the land-use impacts (**Fig 8A**). Results showed that the first principal component (PCA-1) in conjunction with the second principal component (PCA-2) significantly accounted for 84.4% of the total variations of the soil carbon pools and associated parameters as influenced by land-use change. It was observed that soil variables such as CPI, humin, TOC, $FA_{NH}$, NH, FA, and TN were very strongly associated within the CRP and Forest systems when compared to other land-use systems. Even their positive correlations to each other showed a similar pattern when included with soil depths (not shown). The parameters such TN and NPI were also associated with both CRP and Forest. While AC and pb were associated with Bioenergy, the HA was associated with both Wetland and Agriculture; however, other soil parameters did not associate closely with Wetland.

The RDA of the relationship between measured soil parameters and land-use suggested that those soil properties having longer projections (red marked) significantly influenced by

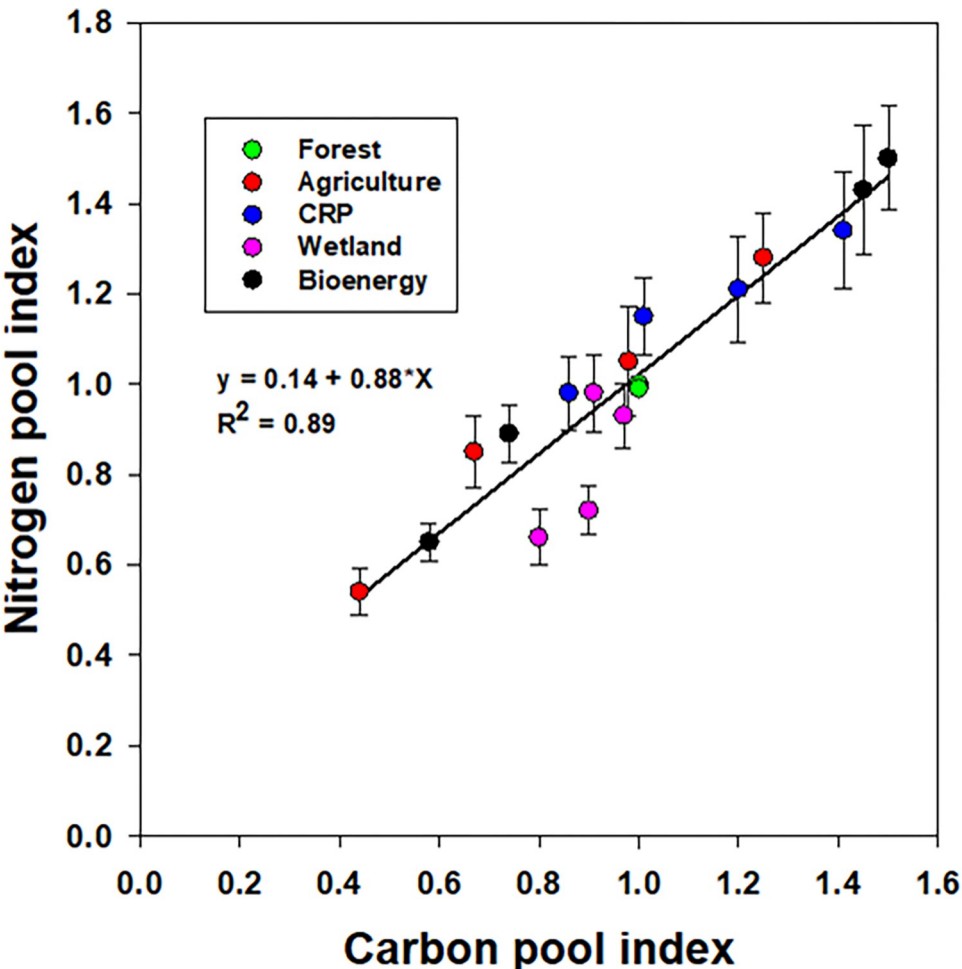

**Fig 7. Relationship between total organic carbon and nitrogen pool indices upon deforestation and subsequent land-use diversity (data were presented with mean ± standard error).**

land-use systems (**Fig 8B**). The FA, TN, TOC, humin, and CPI were significantly impacted by CRP. The higher values of FA, TN, TOC, humin, and CPI under the CRP were due to synergistic relationships among soil biological, chemical, and biological properties as proactively influenced by long-term undisturbed CRP after converting from conventionally-plowed Agriculture. The HA was mostly arising from the soils under both Forest and Bioenergy management systems that may have increased from greater surface deposition of unfragmented residues with high lignin compounds and subsequent slow decomposition associated with HA production. The NPI was correlated to Agriculture in response to the greater availability of N resulting from chemical fertilization to support for crop production. Wetland influenced the ρb positively, due to greater accumulation of TOC, clay, and fine silt materials from edge-of-field loss.

## Conclusions

Converting primary forest resulted in a significant loss in SOC content and lability, particularly under agriculture. However, transitioning from agriculture to the CRP, increased both SOC and TN contents and SOC lability. Agriculture also decreased both FA and NH contents

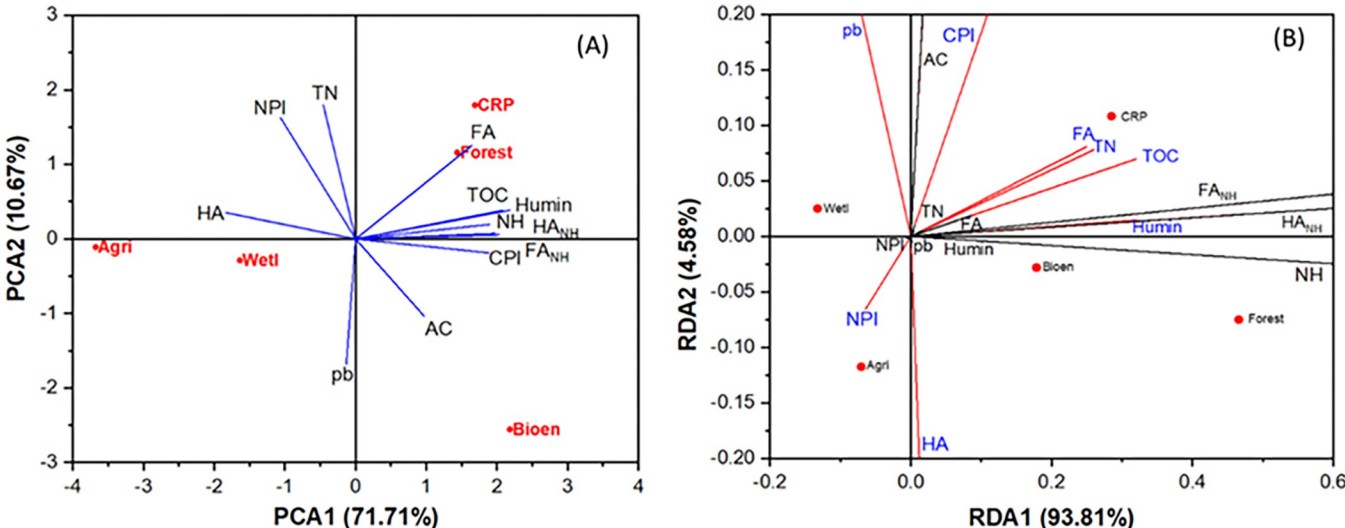

**Fig 8. Principal components analysis (a) and redundancy analysis (RDA) on 12 soil parameters as influenced by land-use management systems [Agri: Agriculture; Bioen: Bioenergy; CRP: Conservation Research program; and Forest, Wetl: Wetland].** Soil parameters: AC is active carbon, CPI is carbon pool index, FA is fulvic acid, FANH is fulvic acid associated non-humic carbon, HA is humic acid, HANH is humic acid associated non-humic carbon, NH is glucose equivalent non-humic carbon, NPI is nitrogen pool index, ρb is soil bulk density, and TN is total nitrogen.

while proportionally increasing HA content in SOC. It was evident from our study that annual plowing potentially leads to a net loss of SOC and its quality and lability, and subsequently, low soil quality. In contrast, an opposite pattern was observed under the CRP. Both SOC and TN pools were more stratified under the CRP when compared to forest as a control. Significantly higher HI and HR values under agriculture with respect to other land-use systems have suggested a more humified SOC accumulation, with an associated depletion of labile C especially FA and $FA_{NH}$ pools. Spectroscopic analyses of $E_4$: $E_6$ ratio suggested an accumulation of SOC that had high degrees of aromaticity but lower proportions of aliphatic structures under agriculture. A significant linear relationship between the CPI and NPI indicated a proportionally more SOC sequestration or depletion with respect to TN accumulation or depletion by land-use change. Moreover, the CMI indicated the positive impact of the CRP management on labile C accumulation in contrast to agriculture, relative to the forest. Results suggested that the CRP management may have favored SOC accumulation with a higher proportion of labile and aliphatic nature than agriculture which may have resulted a greater depletion of labile C with a higher proportion of aromatic compounds in SOC upon deforestation.

## Supporting information

**S1 Dataset.**
(XLSX)

## Acknowledgments

Field work and analysis were conducted at the Soil, Water, and Bioenergy Resources Program of The Ohio State University South Centers at Piketon, Ohio, USA.

## Author Contributions

**Conceptualization:** Emmanuel Amoakwah, Khandakar Rafiq Islam.

**Data curation:** Emmanuel Amoakwah, Mohammad A. Rahman, Khandakar Rafiq Islam.

**Formal analysis:** Emmanuel Amoakwah, Nataliia A. Didenko, Mohammad A. Rahman, Khandakar Rafiq Islam.

**Funding acquisition:** Emmanuel Amoakwah, Khandakar Rafiq Islam.

**Investigation:** Emmanuel Amoakwah, Nataliia A. Didenko, Khandakar Rafiq Islam.

**Methodology:** Emmanuel Amoakwah, Shawn T. Lucas, Nataliia A. Didenko, Khandakar Rafiq Islam.

**Project administration:** Khandakar Rafiq Islam.

**Resources:** Shawn T. Lucas, Mohammad A. Rahman, Khandakar Rafiq Islam.

**Software:** Nataliia A. Didenko, Mohammad A. Rahman, Khandakar Rafiq Islam.

**Supervision:** Khandakar Rafiq Islam.

**Validation:** Emmanuel Amoakwah, Mohammad A. Rahman, Khandakar Rafiq Islam.

**Visualization:** Emmanuel Amoakwah, Shawn T. Lucas, Mohammad A. Rahman, Khandakar Rafiq Islam.

**Writing – original draft:** Emmanuel Amoakwah, Shawn T. Lucas, Nataliia A. Didenko, Khandakar Rafiq Islam.

**Writing – review & editing:** Emmanuel Amoakwah, Nataliia A. Didenko, Khandakar Rafiq Islam.

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
