## [Decision Letter · Decision Letter 0]

28 Feb 2022

PONE-D-22-01007Impact of deforestation and temporal land-use change on soil organic carbon dynamics in a temperate ecosystemPLOS ONE

Dear Dr. Amoakwah,

Thank you for submitting your manuscript to PLOS ONE. After careful consideration, we feel that it has merit but does not fully meet PLOS ONE’s publication criteria as it currently stands. Therefore, we invite you to submit a revised version of the manuscript that addresses the points raised during the review process.

 **The reviewers provided positive comments and I agree with them. Please see the additional editor comments section for further advices on the statistical analysis.** Please submit your revised manuscript by Apr 14 2022 11:59PM. If you will need more time than this to complete your revisions, please reply to this message or contact the journal office at plosone@plos.org. Please include the following items when submitting your revised manuscript:A rebuttal letter that responds to each point raised by the academic editor and reviewer(s). You should upload this letter as a separate file labeled 'Response to Reviewers'.A marked-up copy of your manuscript that highlights changes made to the original version. You should upload this as a separate file labeled 'Revised Manuscript with Track Changes'.An unmarked version of your revised paper without tracked changes. You should upload this as a separate file labeled 'Manuscript'.If applicable, we recommend that you deposit your laboratory protocols in protocols.io to enhance the reproducibility of your results. Protocols.io assigns your protocol its own identifier (DOI) so that it can be cited independently in the future. For instructions see: https://journals.plos.org/plosone/s/submission-guidelines#loc-laboratory-protocols. Additionally, PLOS ONE offers an option for publishing peer-reviewed Lab Protocol articles, which describe protocols hosted on protocols.io. Read more information on sharing protocols at https://plos.org/protocols?utm_medium=editorial-email&utm_source=authorletters&utm_campaign=protocols.

We look forward to receiving your revised manuscript.

Kind regards,

Sergio Saia, Ph.D.

Academic Editor

PLOS ONE

Journal Requirements:

(This work was supported by the Borlaug Leadership Enhancement in Agriculture Program (Borlaug LEAP) through a grant to the University of California-Davis by the United States Agency for International Development (USAID) for professional development of Dr E.A. and Dr. N.D.

The  funders did not play any role in the study design, data collection and analysis, decision to publish, or preparation of the manuscript.)

5. Please note that in order to use the direct billing option the corresponding author must be affiliated with the chosen institute. Please either amend your manuscript to change the affiliation or corresponding author, or email us at plosone@plos.org with a request to remove this option.

6. Please amend the manuscript submission data (via Edit Submission) to include author Khandakar R. Islam.

7. Thank you for stating the following in the Acknowledgments Section of your manuscript: 

(This work was supported by the Borlaug Leadership Enhancement in Agriculture Program (Borlaug LEAP) through a grant to the University of California-Davis by the United States Agency for International Development (USAID) for professional development of Drs. Emmanuel Amoakwah and Nataliia Didenko.)

(This work was supported by the Borlaug Leadership Enhancement in Agriculture Program (Borlaug LEAP) through a grant to the University of California-Davis by the United States Agency for International Development (USAID) for professional development of Dr E.A. and Dr. N.D.

The  funders did not play any role in the study design, data collection and analysis, decision to publish, or preparation of the manuscript.)

8. We note that you have indicated that data from this study are available upon request. PLOS only allows data to be available upon request if there are legal or ethical restrictions on sharing data publicly. For more information on unacceptable data access restrictions, please see http://journals.plos.org/plosone/s/data-availability#loc-unacceptable-data-access-restrictions. 

9. We note that Figure 1 in your submission contain satellite images which may be copyrighted. All PLOS content is published under the Creative Commons Attribution License (CC BY 4.0), which means that the manuscript, images, and Supporting Information files will be freely available online, and any third party is permitted to access, download, copy, distribute, and use these materials in any way, even commercially, with proper attribution. For these reasons, we cannot publish previously copyrighted maps or satellite images created using proprietary data, such as Google software (Google Maps, Street View, and Earth). For more information, see our copyright guidelines: http://journals.plos.org/plosone/s/licenses-and-copyright.

Additional Editor Comments:

please consider changing the figure 1 adding the variable comparisons within measurements and not, as presently shown, the measurements close each other within each land use level. The reader needs to pay attection to the comparison per variable among treatment, not per treatment. Consider, if you agree, also using a multivariate analisis on the standardised variables you measured. Since you have replicates, I suggest the use of a canonical discriminant analysis (please mind to standardise variables before running it). Since you use SAS, please note that the SAS procedure for CDA is Proc Candisc.

Reviewers' comments:

Reviewer's Responses to Questions

**Comments to the Author**

1. Is the manuscript technically sound, and do the data support the conclusions?

Reviewer #1: Yes

Reviewer #2: Yes

2. Has the statistical analysis been performed appropriately and rigorously? 

Reviewer #1: Yes

Reviewer #2: Yes

3. Have the authors made all data underlying the findings in their manuscript fully available?

Reviewer #1: Yes

Reviewer #2: Yes

4. Is the manuscript presented in an intelligible fashion and written in standard English?

Reviewer #1: Yes

Reviewer #2: Yes

5. Review Comments to the Author

Reviewer #1: Dear Authors

General

I read with interest the manuscript entitled “Impact of deforestation and temporal land-use change on soil organic carbon dynamic in a temperate ecosystem”

I think it falls in the aim and scopes of the journal

From my point of view the title did not reflect the specific application, being very broad, it is not addressing the reader to the real topic of the manuscript which is a experimental trial with natural (forest)+specific perennial field crop, conventional tillage (please specify in the abstract).

E 4 : E 6 add ratio

The abstract is confusing, too much acronyms, no clear picture can be drawn. Please spend some time to tighten up the message.

The study area is quite peculiar, I would like to see clearly the site effect and what will change if the catchment-area will be different.

Language has some minor flaws (that could be addressed with proof reading), while few typos are due to inaccuracies in preparing the submission, most of them avoidable.

Introduction

Too broad in general,

The introduction could be improved by looking at a scientific database like SCOPUS and perform a simple search to collect all studies about specific species used in the trial and similar environments,

Avoid groups of references [maximum 2] if 3 please take a moment to explain what was so important and relevant to describe the third and so forth…

Line 139 impact

Line 144 the word stratification is ambiguous, what would you like to say? Stratification is not the contrary of depletion, perhaps sequestration?

Here and there are many terminology inaccuracies that needs revision

Material and methods

Please link the place in the title

Line 289 In the concentration of...(please specify)

Line 293-294 is the LSD the best procedure to assess differences?

Results

fair

Discussion

Please try to stress the relevant point, especially what can be done to ameliorate soils

Conclusion

try to give a take home message first

Line 746, why “temporal”

Kind regards

Suggested reads

For a future more effective literature analysis, to remark the possible publication gap when sourcing only from one citations and abstracts database (e.g SCOPUS), https://www.rendicontionline.it/297/article-4021/modelling-of-soil-organic-carbon-in-the-mediterranean-area-a-systematic-map.html

Reviewer #2: Manuscript Number: PONE- D-22-01007“ Impact of deforestation and temporal land-use change on soil organic carbon dynamics in a temperate ecosystem”

The manuscript addresses a subject of current interest related to the effect of temporal land-use change, from forest to agriculture, wetland, CRP on Soil organic carbon, and total nitrogen Stocks in the temperate climate. Such information in temperate regions is scarce compared to the tropical regions. The novelty of this research, compared to other research when studying the spatial or temporal change of land on SOC and TN stock, is the advanced analysis of carbon and nitrogen pools, namely the humic substances (humic and fulvic acids, humin), the active carbon, the use of humification index (HI), humification ratio (HR), degree of humification (DH), carbon pool index (CPI), nitrogen pool index (NPI), and carbon management index (CMI) to better assess the impacts.

Overall I feel the study can be suitable for publication in PLOS ONE after minor revision. Some suggestions for further improvement are given below :

Page 10 Line 232: please define the parameters and their unit used to calculate C or N Stocks.

Page 18 Line 464: ”Plowing promotes nutrient mineralization under warmer and oxic soil conditions”: Please correct the grammar mistake

Page 22 Line 572: ”the TN and humin stratification patterns we observed were similar to that of SOC.” Please reformulate the sentence.

The quality of the Figures 1, 2, 3, 4, 5, and 6 is too low, it is not possible to read them, Please improve the quality of these figures.

Looking forward to your positive consideration of these comments.

Regards,

Reviewer of the manuscript

6. PLOS authors have the option to publish the peer review history of their article (what does this mean?). If published, this will include your full peer review and any attached files.

Reviewer #1: **Yes: **Calogero Schillaci

Reviewer #2: No

---

## [Author Response · Author response to Decision Letter 0]

5 May 2022

Response to Editor’s comments

Editor’s Comments:

Comment 1: 

Please consider changing the figure 1 adding the variable comparisons within measurements and not, as presently shown, the measurements close each other within each land use level. The reader needs to pay attention to the comparison per variable among treatment, not per treatment. 

Answer:

Thanks. The purpose of showing figure 1 is to just show the chronology of events relative to the conversion of the native (primary) forest to the other land use types. Moreover, the drone-captured picture associated with the Figure 1 shows a pictorial overview of the existing land use types in the study area. Adding the variables considered in this study to figure 1 would make it clumsy and potentially defeat the purpose with which the figure is shown in the manuscript. Moreover, the existing land use picture in Figure 1 was captured by our DJI Mini 2 drone, which belongs to our program at The Ohio State University. So, there is no questions about downloading the picture from the website.

Comment 2:

Consider, if you agree, also using a multivariate analysis on the standardised variables you measured. Since you have replicates, I suggest the use of a canonical discriminant analysis (please mind to standardise variables before running it). Since you use SAS, please note that the SAS procedure for CDA is Proc Candisc.

Answer:

Thanks. As per your suggestion, we run the CDA (SAS and Orion), using all our replicated data including land use and soil depth; however, the results and graphical display were not understandable and visualized to our knowledge (CDA graph attached here). 

Alternatively, we have used principal components (PCA) and redundancy analyses (RDA), which show satisfactory results and proper visualization of the relationships among the variables (added within the text of the manuscript. We have followed figures and graphs in few of your papers. I hope the PCA, and RDA will address the issues.

Response to Reviewers

REVIEWER #1

General

I read with interest the manuscript entitled “Impact of deforestation and temporal land-use change on soil organic carbon dynamic in a temperate ecosystem”. I think it falls in the aim and scopes of the journal.

From my point of view the title did not reflect the specific application, being very broad, it is not addressing the reader to the real topic of the manuscript which is an experimental trial with natural (forest)+specific perennial field crop, conventional tillage (please specify in the abstract).

Specific comments:

Comment/Suggestion Response

E 4: E 6 add ratio 

Response: Per suggestion, the revisions were made thoroughly within the text of the manuscript. Please see lines42, 50, 51, 64, 585, 586, 588, 663, 690, 693, 695, 696, 756.

The abstract is confusing, too much acronyms, no clear picture can be drawn. Please spend some time to tighten up the message. 

Response: The abstract was edited and concise. Few of the acronyms were taken out and all of them have been defined. A table (Table 1) was included in the M&M section to explain all the acronyms.

The study area is quite peculiar, I would like to see clearly the site effect and what will change if the catchment-area will be different. 

Response: The study area consists of all the sites on a relatively flat and homogeneous land except the differences in management systems over time. That is what we tried to show with regards to how management systems over time changed the site quality in terms of changes in soil organic carbon contents, quality, and lability, including other soil properties. If the catchment area were different from the area where this study was done, there could be differences with regards to the site effect on the measured variables. Depending on the soil properties of the catchment area, land use type following the conversion of the primary forest, the level or amount of input used coupled with the magnitude of disturbance of the soil ecosystem, the carbon and nitrogen stoichiometry could be affected. 

Language has some minor flaws (that could be addressed with proof reading), while few typos are due to inaccuracies in preparing the submission, most of them avoidable. 

Response: The manuscript has gone through a proof reading as much as we could do, and typo errors and grammatical inaccuracies have been corrected accordingly.

Introduction: Too broad in general,

The introduction could be improved by looking at a scientific database like SCOPUS and perform a simple search to collect all studies about specific species used in the trial and similar environments 

Response: Thank you for your comment. The scope to this study was to understand the SOC pools and their lability, and how the SOC pools are impacted by land-use change to provide useful information on the land management implications for enhanced soil ecosystem functions and services. In as much as we appreciate the importance of collecting all studies on plant species studied under various ecosystems in similar temperate environments, we respectfully believe that it would not fit the scope of this study. As suggested, few relevant literatures were cited to focus on the studies conducted under similar climates.

Avoid groups of references [maximum 2] if 3 please take a moment to explain what was so important and relevant to describe the third and so forth… 

Response: In all, groups of references have been reduced to a maximum of two, where applicable. Only in few cases, three references were cited to link and justify the results as there were updated terms and calculations used in this manuscript.

Line 139 impact Revised, as suggested. 

Response: Please see line 137.

Line 144 the word stratification is ambiguous, what would you like to say? Stratification is not the contrary of depletion, perhaps sequestration? 

Response: The word “stratification” used in this context is not ambiguous. Stratification is contrary to overall sequestration or depletion in response to the variable management systems and inputs when compared to the baseline or control systems deeper soil depth. It is not necessarily synonymous to sequestration, but it is an indicator of sequestration and defines the soil ecosystem functions and services (Franzluebbers et al. 2002).

Here and there are many terminology inaccuracies that needs revision 

Response: All the terminologies have been well defined. 

Material and methods

Please link the place in the title 

Response: Thank you for the suggestion. The title of the manuscript is simplified, as suggested. 

Line 289 In the concentration of...(please specify) 

Response: The sentence is unambiguously clear. We looked at the concentrations (%, mg/kg, g/kg), stocks (Mg/ha), and stratification (unitless ratio) of soil organic carbon and total nitrogen pools. “Significant differences in the concentration, stocks, and stratification of SOC and TN pools…” Please see lines 287 – 289.

Line 293-294 is the LSD the best procedure to assess differences? 

Response: There are other statistical procedures to assess statistical differences between means in experimental trials. In this study, the widely used “Least Significant Difference (LSD) Test” was considered appropriate to elucidate the interactive effects of the predictor variables on dependent variables and to assess the mean differences between land use types relative to the respective measured variables. Moreover, principal components- and redundancy analyses (PCA and RDA) were performed to improve the statistical validity and transferability of the results.

Results: fair 

Response: Revisions were made to improve flow and quality of the presentation.

Discussion:

Please try to stress the relevant point, especially what can be done to ameliorate soils 

Response: Thanks. We focused on the evaluation of how the land use affected soil carbon pools and TN content over time. The CRP and other conservation management practices (such as establishment of bioenergy plantations under no-till) can be utilized to ameliorate the degraded and/or disturbed ecosystems with especial reference to soil quality and ecosystem services. More information was added in the manuscript. Please see lines 450 – 455.

Conclusion:

Try to give a take home message first 

Response: A take home message “stop deforestation and follow conservation practices such as CRP to sequester C and mitigate climate change effects” has been incorporated into the conclusion section of the manuscript. Please see lines 750 – 751; 761 – 764.

Line 746, why “temporal” 

Response: The word “temporal” has been deleted from the sentence. Please see line 747.

REVIEWER #2

The manuscript addresses a subject of current interest related to the effect of temporal land-use change, from forest to agriculture, wetland, CRP on Soil organic carbon, and total nitrogen Stocks in the temperate climate. Such information in temperate regions is scarce compared to the tropical regions. The novelty of this research, compared to other research when studying the spatial or temporal change of land on SOC and TN stock, is the advanced analysis of carbon and nitrogen pools, namely the humic substances (humic and fulvic acids, humin), the active carbon, the use of humification index (HI), humification ratio (HR), degree of humification (DH), carbon pool index (CPI), nitrogen pool index (NPI), and carbon management index (CMI) to better assess the impacts. Overall, I feel the study can be suitable for publication in PLOS ONE after minor revision. 

Some suggestions for further improvement are given below:

Comment/Suggestion Response +Page/Line numbers

Page 10 Line 232: please define the parameters and their unit used to calculate C or N Stocks. 

Response: Thanks. Revisions were performed, as suggested through-out the manuscript. Please see lines 206 – 207.

Page 18 Line 464: ”Plowing promotes nutrient mineralization under warmer and oxic soil conditions”: Please correct the grammar mistake 

Response: The sentence has been revised. Please see Lines 445 – 448.

Page 22 Line 572: ”the TN and humin stratification patterns we observed were similar to that of SOC.” Please reformulate the sentence. 

Response: The sentence is reformulated to make it comprehensible. Please see Line 554.

The quality of the Figures 1, 2, 3, 4, 5, and 6 is too low, it is not possible to read them, please improve the quality of these figures. 

Response: Thank you for your suggestion. The quality of the figures has been improved upon.

---

## [Editor Report · Decision Letter 1]

30 May 2022

Impact of deforestation and temporal land-use change on soil organic carbon storage, quality, and lability

PONE-D-22-01007R1

Dear Dr. Amoakwah,

We’re pleased to inform you that your manuscript has been judged scientifically suitable for publication and will be formally accepted for publication once it meets all outstanding technical requirements.

Kind regards,

Sergio Saia, Ph.D.

Academic Editor

PLOS ONE
---

## [Editor Report · Acceptance letter]

19 Jul 2022

PONE-D-22-01007R1 

Impact of deforestation and temporal land-use change on soil organic carbon storage, quality, and lability 

Dear Dr. Islam:

I'm pleased to inform you that your manuscript has been deemed suitable for publication in PLOS ONE. Congratulations! Your manuscript is now with our production department. 

Kind regards, 

on behalf of

prof Sergio Saia 

Academic Editor

PLOS ONE